# The mechanisms to dispose of misfolded proteins in the endoplasmic reticulum of adipocytes

Shuangcheng Alivia Wu[1], Chenchen Shen[2], Xiaoqiong Wei ®[1], Xiawei Zhang[1], Siwen Wang[1], Xinxin Chen[1], Mauricio Torres[1], You Lu ®[1], Liangguang Leo Lin[1], Huilun Helen Wang[1], Allen H. Hunter[3], Deyu Fang ®[4], Shengyi Sun[5], Magdalena I. Ivanova[6,7], Yi Lin[2] ✉ & Ling Qi ®[1,8] ✉

Endoplasmic reticulum (ER)-associated degradation (ERAD) and ER-phagy are two principal degradative mechanisms for ER proteins and aggregates, respectively; however, the crosstalk between these two pathways under physiological settings remains unexplored. Using adipocytes as a model system, here we report that SEL1L-HRD1 protein complex of ERAD degrades misfolded ER proteins and limits ER-phagy and that, only when SEL1L-HRD1 ERAD is impaired, the ER becomes fragmented and cleared by ER-phagy. When both are compromised, ER fragments containing misfolded proteins spatially coalesce into a distinct architecture termed Coalescence of ER Fragments (CERFs), consisted of lipoprotein lipase (LPL, a key lipolytic enzyme and an endogenous SEL1L-HRD1 substrate) and certain ER chaperones. CERFs enlarge and become increasingly insoluble with age. Finally, we reconstitute the CERFs through LPL and BiP phase separation in vitro, a process influenced by both redox environment and C-terminal tryptophan loop of LPL. Hence, our findings demonstrate a sequence of events centered around SEL1L-HRD1 ERAD to dispose of misfolded proteins in the ER of adipocytes, highlighting the profound cellular adaptability to misfolded proteins in the ER in vivo.

Cells have evolved various quality control mechanisms to maintain proteostasis in various organelles to prevent the formation of potentially toxic protein aggregates[1,2]. The collapse of protein quality control system(s) may indeed contribute to disease pathogenesis such as in aging and diabetes[1]. In the endoplasmic reticulum (ER), misfolded proteins are cleared by two degradative mechanisms, namely ER-associated degradation (ERAD) and selective macroautophagy known as ER-phagy[3]. While ERAD degrades proteins in a substrate-specific manner[4–8], ER-phagy represents a mechanism for lysosome-mediated bulk clearance of ER fragments or subdomains. Recent studies have demonstrated the importance of ER-phagy in vitro toward protein aggregates such as disease-relevant mutant proteins of proinsulin[9], procollagen[10,11], POMC[12,13], and NPC1[14]. However, while it has been reported that ER-phagy degrades ERAD-resistant model substrates

[1]Department of Molecular & Integrative Physiology, University of Michigan Medical School, Ann Arbor, MI 48105, USA. [2]Tsinghua-Peking Center for Life Science, IDG/McGovern Institute for Brain Research, School of Life Sciences, Tsinghua University, Beijing 100084, China. [3]College of Engineering and Michigan Center for Materials Characterization, University of Michigan, Ann Arbor, MI 48109, USA. [4]Department of Pathology, Northwestern University Feinberg School of Medicine, Chicago, IL 60611, USA. [5]Center for Molecular Medicine and Genetics, Department of Biochemistry, Microbiology and Immunology, Wayne State University School of Medicine, Detroit, MI 48201, USA. [6]Department of Neurology, University of Michigan, Ann Arbor, MI 48109-5622, USA. [7]Biophysics Program, University of Michigan, Ann Arbor, MI, USA. [8]Division of Metabolism, Endocrinology & Diabetes, Department of Internal Medicine, University of Michigan Medical School, Ann Arbor, MI 48105, USA. ✉e-mail: linyi@mail.tsinghua.edu.cn; lingq@umich.edu

in vitro[15], the physiological relevance and significance of ER-phagy and its crosstalk with ERAD in vivo remain largely unclear[16–19].

The ER-resident SEL1L-HRD1 protein complex (Hrd3p-Hrd1p in yeast) represents the most conserved branch of ERAD[4,5,20–23], where SEL1L not only controls the stability of the E3 ligase HRD1[20–24], but defines substrate specificity and HRD1 function[24,25]. Loss of SEL1L or HRD1, either in germline or acutely in adult mice, cause lethality[24,26–28], highlighting the importance of SEL1L-HRD1 ERAD in organismal development and function. Indeed, using cell type-specific mouse models, we and others have shown that SEL1L-HRD1 ERAD is indispensable for fundamental physiological processes such as immune cell development and function, gut homeostasis, water balance, food intake, systemic energy homeostasis, nutrient metabolism and organellar crosstalk in a substrate-dependent manner[5,20,24,29–41]. Peculiarly, in most cell types examined to date, SEL1L-HRD1 ERAD deficiency is coupled with only very mild ER stress or unfolded protein response (UPR), with no detectable cell death before, at or even after disease initiation in vivo[4,5,20], pointing to the existence of compensatory mechanism(s) in response to impaired SEL1L-HRD1 ERAD function[42].

Using adipocyte-specific genetic knockout (KO) mouse models powered by a combination of state-of-the-art ultrastructural imaging, proteomic screens, biophysical and biochemical assays, we uncover the sequence of events underlying the intimate crosstalk between SEL1L-HRD1 ERAD and ER-phagy in adipocytes in vivo, to dispose of a misfolded proteins in the ER, such as lipoprotein lipase (LPL). In the absence of SEL1L-HRD1 ERAD, autophagy becomes active to clear ER fragments containing misfolded proteins and/or aggregates. In adipocytes lacking both SEL1L-HRD1 ERAD and autophagy, these ER fragments are coalesced into a unique cellular compartment termed CERFs. LPL, a key enzyme involved in systemic lipid metabolism and partitioning[43], and ER chaperone BiP are two main components of CERFs and undergo phase separation in vitro.

## Results

### Activation of autophagy in adipocytes lacking SEL1L
We previously showed that *Sel1L* deficiency in adipocytes (*Sel1L^{AdipCre}*) is associated with very mild UPR activation[31] as measured by *Xbp1* mRNA splicing (Supplementary Fig. 1a), and without cell death as measured by cleaved caspase 3 and TUNEL staining (Supplementary Fig. 1b, c), pointing to the existence of compensatory mechanism(s) in response to impaired SEL1L-HRD1 ERAD function. Indeed, KEGG pathway enrichment analysis of cDNA profiling microarray analysis of white adipose tissues (WAT) (GSE56918)[31] revealed autophagy among the top pathways upregulated in *Sel1L^{AdipCre}* WAT compared to WT WAT (Supplementary Fig. 1d). Indeed, ATG7-dependent conversion of microtubule-associated protein 1 light chain (LC3) LC3-I to LC3-II (LC3-phosphatidylethanolamine conjugate) was doubled in WAT of *Sel1L^{AdipCre}* mice compared to that of WT littermates (Fig. 1a). On the other hand, protein level of the ubiquitin-binding autophagy receptor and substrate P62[44,45] was largely unchanged while highly accumulated in *Atg7^{AdipCre}* WAT (Fig. 1a), excluding the possibility of defective lysosomal function in *Sel1L^{AdipCre}* WAT. Moreover, transmission electron microscopy (TEM) analysis of WAT revealed that there were more double-membrane autophagosomes (arrows) containing ER-like structures (asterisks) in *Sel1L^{AdipCre}* WAT compared to that in WT WAT under basal state (arrows, Fig. 1b, c and Supplementary Fig. 1e), pointing to ER-phagy. Hence, similar to our recent report in pancreatic β cells[42], our data showed an activation of autophagy and ER-phagy in adipocytes lacking SEL1L.

### Generation of mice with adipocyte-specific deletion of *Sel1L* and *Atg7*
To examined the physiological significance of the crosstalk between ERAD and autophagy in adipocytes, we generated a mouse model lacking both *Sel1L* and *Atg7* in adipocytes (*Sel1L^{AdipCre};Atg7^{AdipCre}*,

hereafter *DKO*) using the Adiponectin-Cre mice[31]. Wildtype (WT) and adipocyte-specific *Sel1L* or *Atg7* deficient (*Sel1L^{AdipCre}* or *Atg7^{AdipCre}*) littermates were included as controls. Both genders of adult mice were used throughout this study. Western blot analyses confirmed the reduction of SEL1L, HRD1 or ATG7 protein levels in WAT (Fig. 1d). At 12 weeks of age, *DKO* mice, of both genders, exhibited comparable body weights as the other three cohorts (Fig. 1e). The size and weight of white fat pads were reduced in *DKO* WAT compared to the other cohorts (Fig. 1f, g), which was likely caused by the inability of WAT to expand from 5 to 12 weeks of age (Fig. 1h). Histologically, *DKO* adipocytes exhibited smaller lipid droplets and cell size compared to the other cohorts at 12 weeks of age (Fig. 1i, j). There was no detectable cell death as measured by caspase-3 cleavage and TUNEL staining (Fig. 1j and Supplementary Fig. 2a), while a modest increase of *Xbp1* mRNA splicing (Supplementary Fig. 2b, c), a marker of UPR IRE1α activity[46,47], in *DKO* WAT compared to other cohorts.

### ER fragments spatially coalesce into clusters (CERFs) in *DKO* adipocytes
The forementioned data prompted the question how *DKO* cells deal with misfolded proteins in the ER and cytosol. Loss of autophagy is known to cause the formation of protective P62-positive cytosolic inclusions, containing ubiquitinated proteins and protein aggregates[44], which may include ER fragments and contents as suggested by recent studies[48,49]. Indeed, under transmission electron microscopy (TEM), we noted the presence of the typical P62 inclusions in *DKO* WAT and brown adipose tissues (BAT) (Fig. 2a). This was further confirmed using P62 immunogold labeling followed by TEM (Fig. 2b) and confocal microscopy (Supplementary Fig. 3a). P62-containing inclusions were also detected in *Atg7^{AdipCre}* WAT/BAT, albeit of much smaller size and at much lower frequency than those in *DKO* WAT/BAT (Supplementary Fig. 3a). Upon co-staining with the ER marker KDEL (for two major ER chaperones BiP and GRP94), we noted that KDEL proteins were largely excluded from the P62 inclusions in *DKO* WAT (Fig. 2b). Hence, we concluded that misfolded proteins in the ER are not predominantly shunted to the P62 inclusions.

Excitingly, under immunogold-TEM, we frequently observed clusters of the KDEL signals in addition to the P62 inclusions in the cytosol of *DKO* WAT (green dotted lines, Fig. 2c), which were not present in other cohorts (Supplementary Fig. 3b). These ER clusters had very few detectable P62 signals inside (Fig. 2c), hence representing a cellular compartment distinct from the P62 inclusions. TEM analysis indeed revealed a unique, membranous, cellular compartment in *DKO* WAT, but not in other cohorts (arrows, Fig. 2d and Supplementary Fig. 3c, d). The same structure of clusters of ER fragments was observed in *DKO* BAT as well (Fig. 2e, f). 3D FIB-SEM analysis showed that this architecture was a distinct oval-shaped compartment composed of ER fragments (Fig. 2g and Supplementary Video 1). The diameters of the clusters ranged between 2 and 7 μm, slightly smaller than those of the P62 inclusions in *DKO* WAT (Fig. 2h). Providing further support for the nature of ER fragments, we noted clusters of large ER fragments in *DKO* adipocytes (Fig. 2i), some of which might have fused at the core (arrows, Fig. 2j and Supplementary Video 1). Taken together, these data suggested that ER fragments may coalesce into a unique cellular architecture in adipocytes lacking SEL1L and ATG7, which we termed as the coalescence of ER fragments (CERFs).

### LPL is a misfolded SEL1L-HRD1 ERAD substrate
We next explored the key protein component(s) of CERFs. The observation that CERFs are clusters of ER fragments only in *DKO* cells prompted us to hypothesize that ER fragments contain misfolded ERAD substrates. To this end, we designed and performed an exploratory immunoprecipitation mass spectrometry (IP-MS) experiment (without replication) to identify potential SEL1L-HRD1 ERAD substrates in BAT. As HRD1 deficiency allows

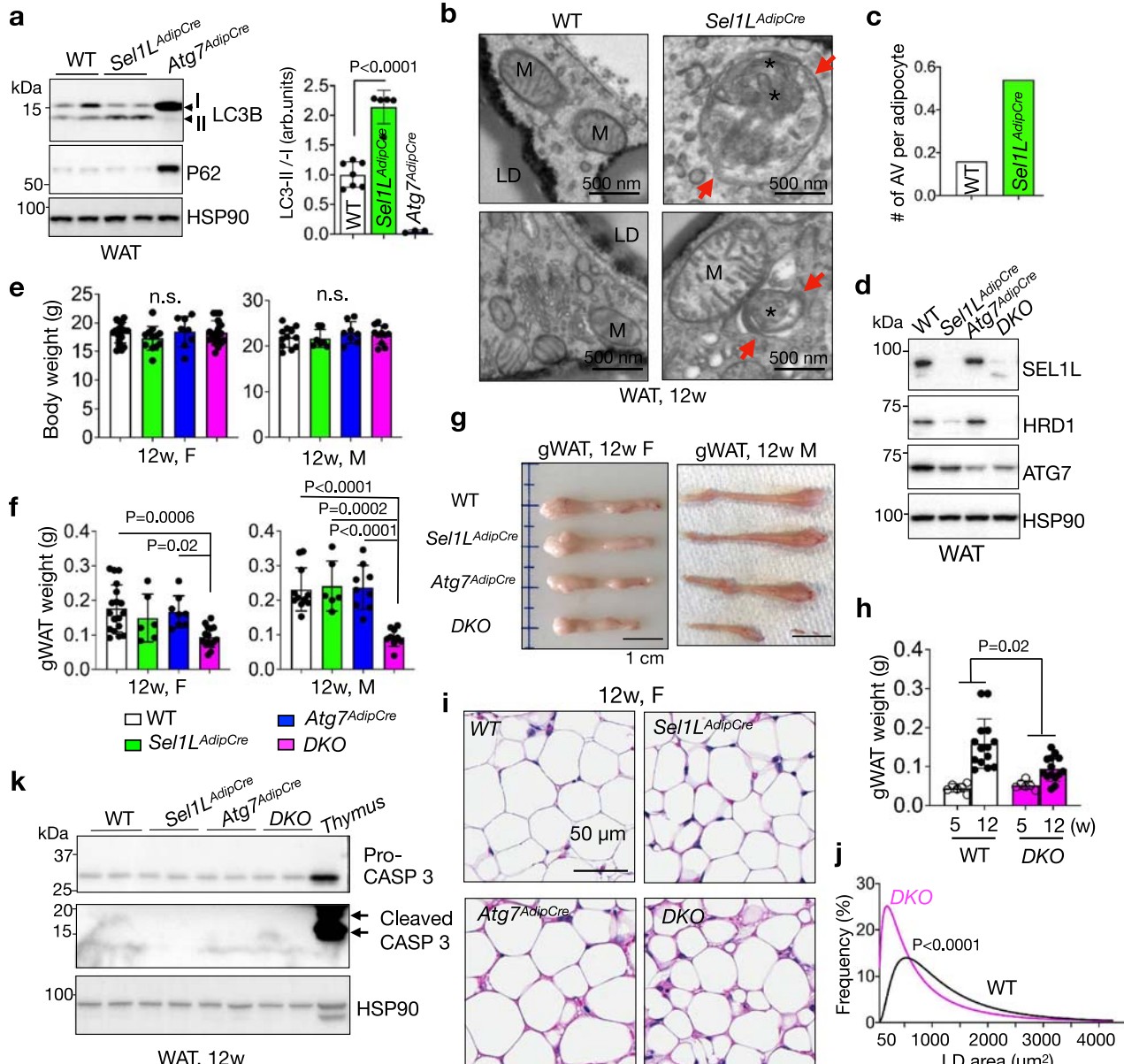

**Fig. 1 | Elevated autophagic activity in Sel1L-deficient adipocytes. a** Immunoblot analysis of LC3B and P62 in gonadal WAT with quantitation of LC3B-II to I conversion ratio shown on the right (*n* = 7 for WT and 5 for *Sel1L*^AdipCre^ mice). **b, c** Representative TEM images of gonadal WAT showing ER-like structures (*) in the autophagosomes (arrow), with quantitation of autophagic vacuoles (AV) shown in (c); n = 50 WT and 63 *Sel1L*^AdipCre^ adipocytes from 3 mice each. LD, lipid droplet; M, mitochondria. **d** Immunoblot analysis of SEL1L-HRD1 ERAD and autophagy in gonadal WAT (n = 4 mice per genotype). **e** Body weight of female (F, n = 19 for WT, 12 for *Sel1L*^AdipCre^, 8 for *Atg7*^AdipCre^ and 17 for *DKO*) and male (M, n = 12 for WT, 7 for *Sel1L*^AdipCre^, 8 for *Atg7*^AdipCre^ and 11 for *DKO*) littermates at the age of 12 weeks. Weight (**f**) and representative pictures (**g**) of gonadal WAT of female (F, n = 18 for WT, 6 for *Sel1L*^AdipCre^, 8 for *Atg7*^AdipCre^ and 15 for *DKO*) and male (M, n = 11 for WT, 6 for *Sel1L*^AdipCre^, 9 for *Atg7*^AdipCre^ and 9 for *DKO*) littermates at 12 weeks of age. **h** Weight of gonadal WAT of female littermates at 5 and 12 weeks of age (n = 6 mice at 5 weeks of age per genotype, n = 14 mice at 12 weeks of age per genotype). **i, j** Representative H&E images of gonadal WAT of female littermates at 12 weeks of age (n = 3 mice per genotype), with quantitation of area of lipid droplets (LD) shown in (**j**) (n = 960 and 1042 cells from two mice per genotype). **k** Immunoblot analysis of (cleaved) caspase 3 in gonadal WAT of 12-week-old mice (n = 4 mice per genotype). Mouse thymus, a positive control. Histogram was plotted as mean with SD; each data points were derived from biologically independent mice. n.s., not significant; P values were derived by two-sided Student's *t*-test (**a**), One-way ANOVA followed by Tukey's test (**e, f**), Two-way ANOVA (**h**) and Chi-square test (**j**). Source data are provided as a Source Data file.

the capture of the interactions between SEL1L and substrates[5,20], we performed SEL1L immunoprecipitation in BAT from mice with brown adipocyte-specific HRD1 deficiency (*Hrd1*^Ucp1Cre^)[39], with WT and *Sel1L*^Ucp1Cre^ BAT as controls (Supplementary Fig. 4a). Only proteins showing 4- and 2-fold increases in peptide-spectrum matches (PSMs) in *Hrd1*^Ucp1Cre^ samples over *Sel1L*^Ucp1Cre^ and WT samples, respectively, were considered as potential substrates. A total of 40 candidates were identified (Supplementary Table 1), which included known SEL1L interactors such as HSP90B1

(GRP94), OS9 and ERLEC1 (XTP3B)[50] (Supplementary Fig. 4b), hence validating our experimental system. Excitingly, lipoprotein lipase (LPL), a rate-limiting enzyme catalyzing the hydrolysis of triglycerides packed in plasma lipoproteins into fatty acids[43], was identified among the top hits (Supplementary Fig. 4b).

We previously showed that LPL is largely trapped in the ER of *Sel1L*-deficient adipocytes[31], but it was unclear whether LPL is an endogenous substrate of SEL1L-HRD1 ERAD. LPL interacted with SEL1L in *Hrd1*^−/−^ but not WT adipocytes (Fig. 3a). Protein levels of LPL were

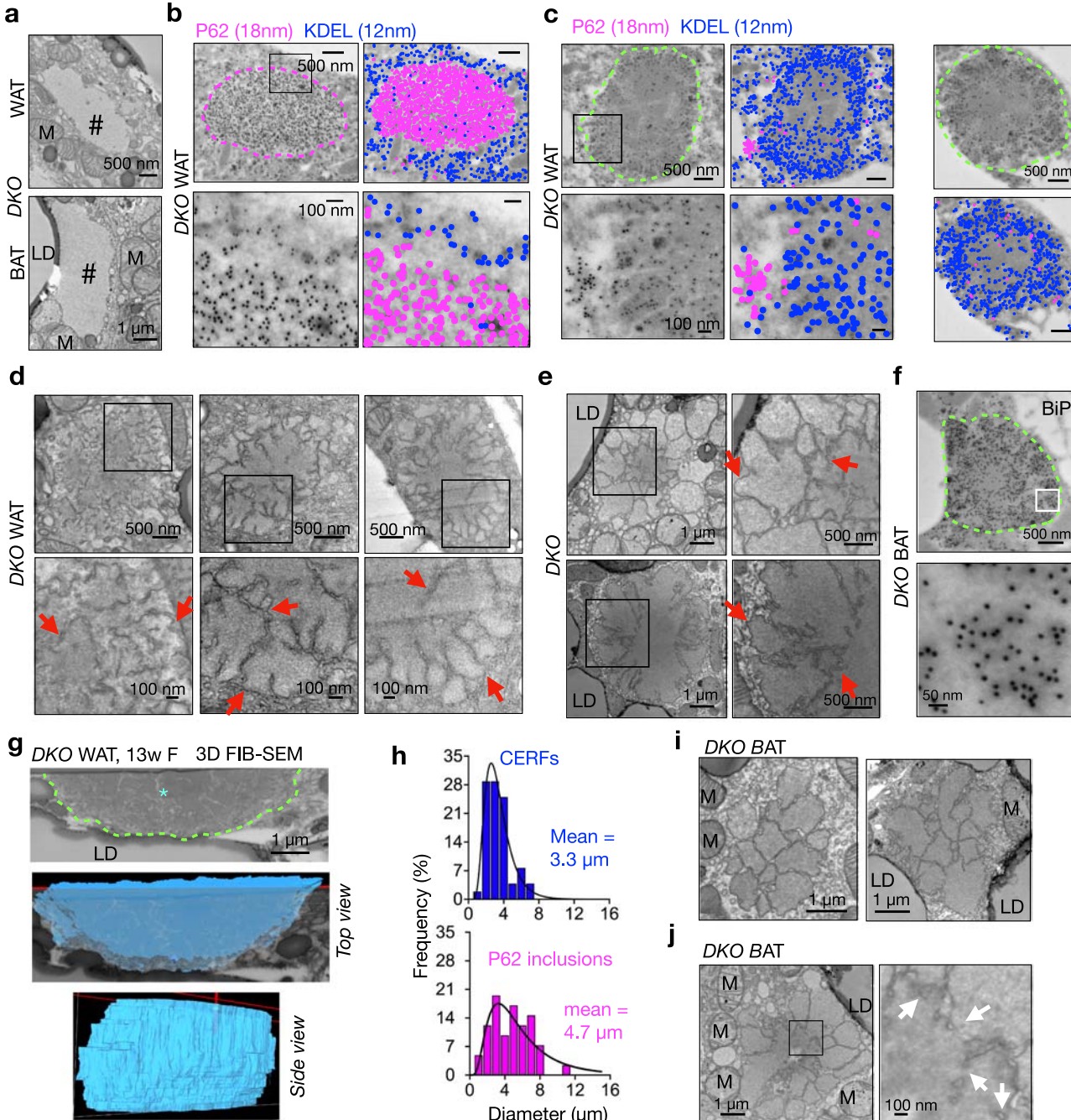

**Fig. 2 | Coalescence of ER fragments (CERFs) in *DKO* adipocytes.**
**a** Representative TEM images of P62 inclusions in *DKO* gonadal WAT (n = 4 mice) and BAT (n = 3 mice). #, P62 inclusion; LD, lipid droplet; M, mitochondria.
**b, c** Representative EM images following immunogold co-labeling of P62 (pink dotted line) and the ER marker KDEL (green dotted line) in *DKO* gonadal WAT (n = 2 mice). Color-coded images, P26 (pink) and KDEL (blue). **d–f** Representative TEM images (**d, e**, n = 3-4 mice each) and TEM with immunogold labeling of the ER

marker BiP (**f**, n = 3 mice) in *DKO* gonadal WAT or BAT. Arrows, membranes. **g** 3D reconstruction of a CERF in *DKO* gonadal WAT from 370 FIB-SEM images taken at every 10 nm (n = 2 independent repeats). **h** Diameters of the CERFs (n = 52) and P62 inclusions (n = 49) in *DKO* gonadal WAT based on TEM analysis of 4 mice with the mean of diameters indicated. **i, j** TEM images showing possible clustering (**i**) and fusion (**j**) of ER fragments in *DKO* BAT (n = 3 mice). Arrows, likely fused membranes inside CERFs. Source data are provided as a Source Data file.

stabilized and accumulated in *Sel1L⁻/⁻* or *Hrd1⁻/⁻* adipocytes versus that in WT adipocytes (Fig. 3b, c). Consistently, LPL accumulated by threefold to fourfold in *Sel1L*-deficient WAT (Fig. 3d, e), suggesting that LPL is indeed a misfolded SEL1L-HRD1 ERAD substrate. Unlike in *Sel1L^{AdipCre}* WAT, *Atg7* deficiency in WAT had a subtle effect on LPL protein level (Fig. 3d, e). Moreover, LPL protein levels were further increased by another ~5-folds in *DKO* WAT over that in *Sel1L^{AdipCre}* WAT (Fig. 3d, e). The accumulation of LPL in *Sel1L^{AdipCre}* or *DKO* WAT was not a general phenomenon for membrane or secretory proteins as others

such as calnexin and glucose transporter 4 (GLUT4) were unchanged compared to those in WT WAT (Fig. 3d, f, g). These changes in LPL protein levels were uncoupled from gene transcription as mRNA level of *Lpl* was comparable among all four cohorts (Fig. 3h).

## LPL is a major component of CERFs
LPL is normally secreted from adipocytes to the capillary where it hydrolyzes triglyceride-rich lipoproteins[43]. Endoglycosidase H (EndoH) digestion showed that over 90% of LPL in *Sel1L^{AdipCre}* and *DKO* WAT was

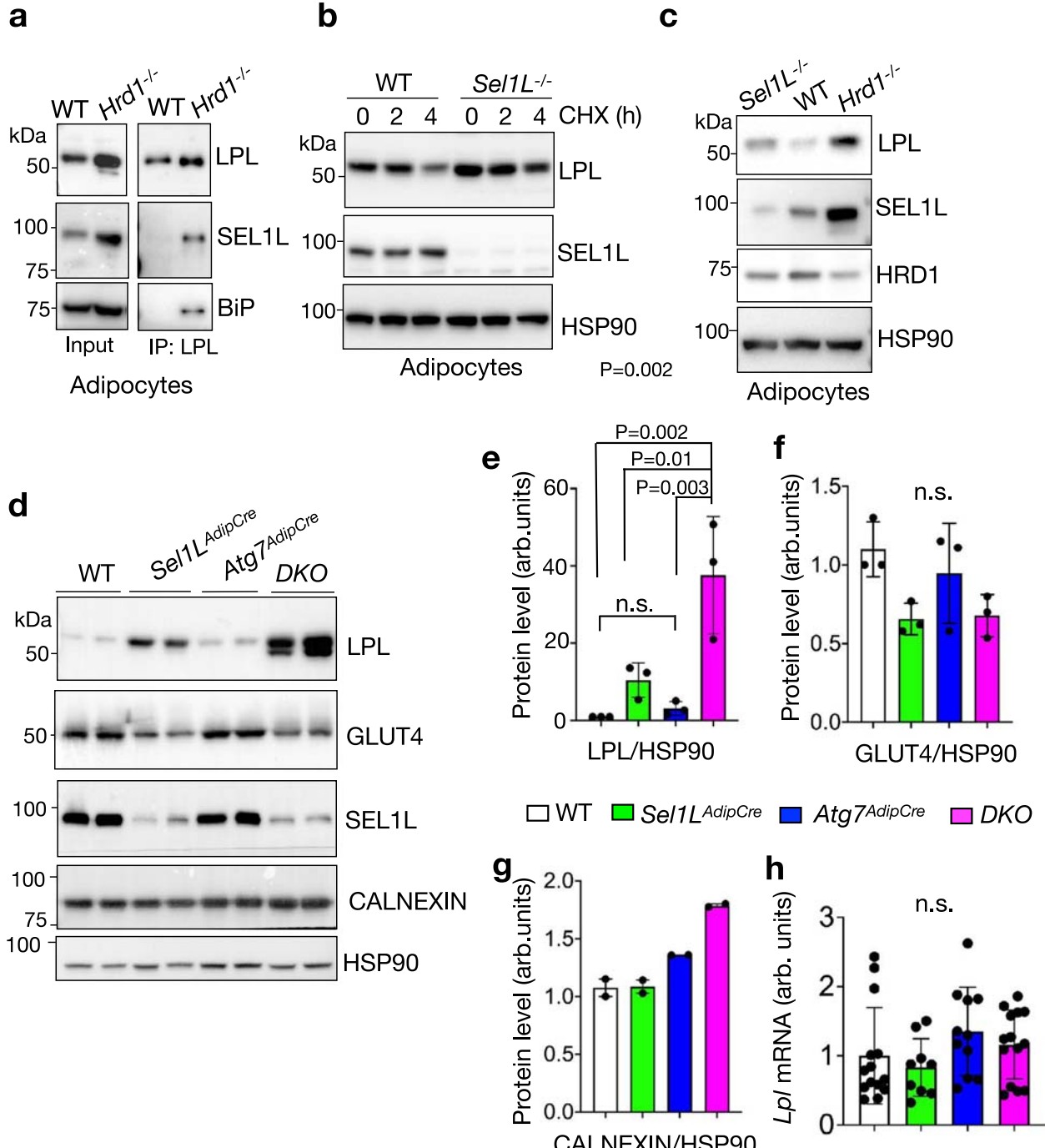

**Fig. 3 | LPL is an endogenous SEL1L-HRD1 ERAD substrate in adipocytes.**
**a** Immunoblot analysis following immunoprecipitation of endogenous LPL in differentiated adipocytes (3 independent repeats). **b** Immunoblot analysis of LPL in differentiated adipocytes pre-treated with Brefeldin A for 30 min followed by cycloheximide (CHX) treatment for the indicated times (2 independent repeats). **c** Immunoblot analysis of LPL in differentiated adipocytes (3 independent repeats). **d–g** Immunoblot analysis of LPL and other proteins in gonadal WAT (d) with

quantitation shown in e-g (n = 3 mice per genotype for LPL and GLUT4 and n = 2 mice per genotype for CALNEXIN). **h** qPCR analysis of *Lpl* mRNA levels in gonadal WAT (n = 14 per WT and *DKO*, n = 9 for *Sel1L^AdipCre* and n = 11 for *Atg7^AdipCre*). Histogram was plotted as mean with SD; each data points were derived from biologically independent mice/samples. P values were derived by one-way ANOVA followed by Tukey's test; n.s. not significant. Source data are provided as a Source Data file.

retained in the ER (i.e. EndoH sensitive), unlike those in WT and *Atg7^AdipCre* WAT with 60% being in extra-ER compartments (i.e. EndoH resistant) (Fig. 4a, b), suggesting that SEL1L is required for the maturation of nascent LPL in the ER. Immunofluorescent staining showed that LPL was trapped intracellularly and largely absent from the endothelium in *Sel1L^AdipCre* and *DKO* WAT (white arrows, Fig. 4c, d and Supplementary Fig. 5a). By contrast, majority of LPL was co-localized

with the capillary marker GPIHBP1 in WT and *Atg7^AdipCre* WAT (yellow arrows, Fig. 4c, d and Supplementary Fig. 5a). In keeping with intracellular retention of LPL, *DKO* mice exhibited mild postprandial hyperlipidemia compared to other cohorts (Supplementary Fig. 5b).

Unlike that in *Sel1L^AdipCre* WAT, LPL protein formed large puncta in *DKO* WAT (green arrows, Fig. 4c, d). These LPL-containing puncta were formed inside of adipocytes as demonstrated by co-staining with a

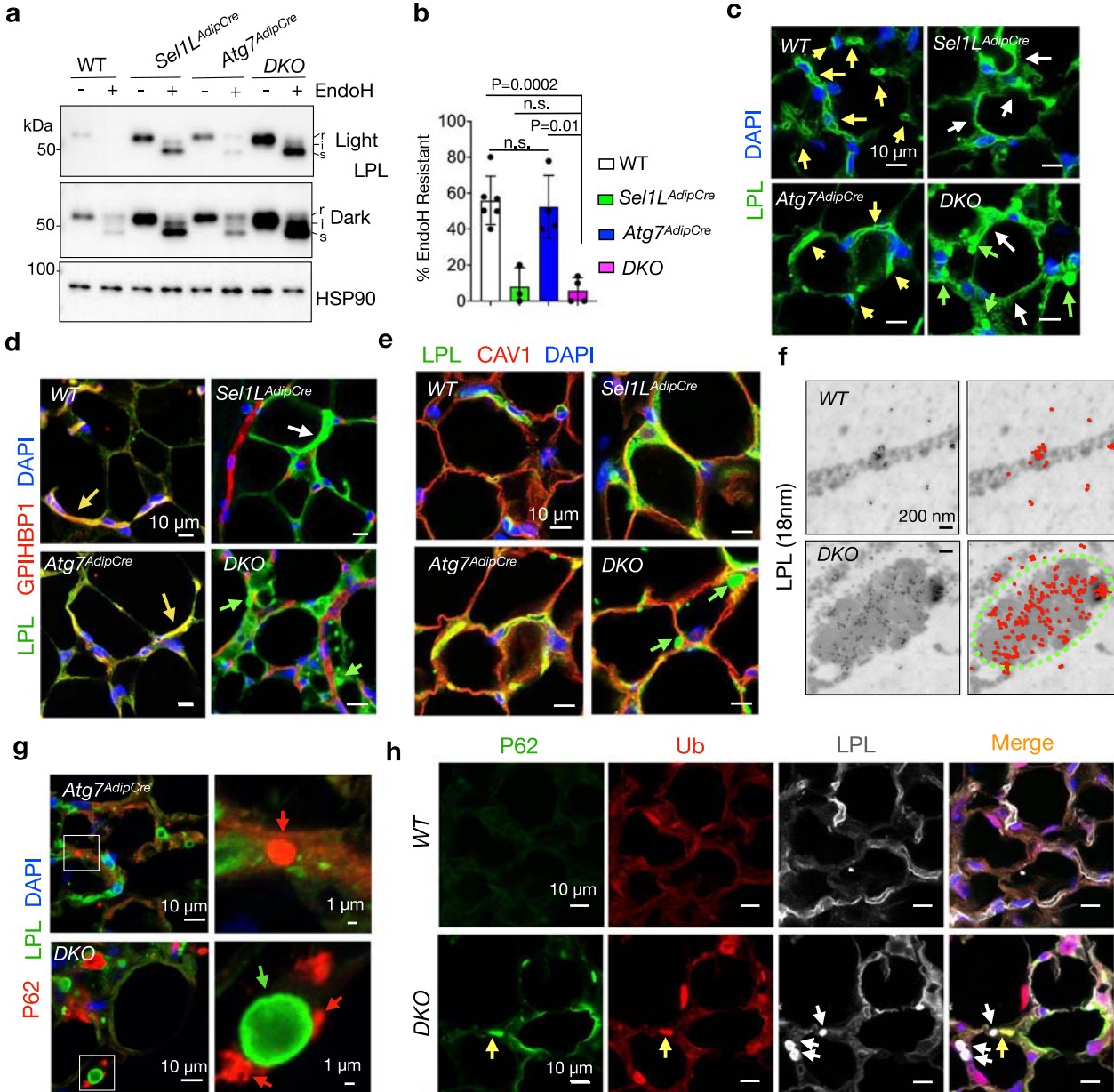

**Fig. 4 | LPL is a principal component of the CERFs. a, b** EndoH analysis of LPL in gonadal WAT using Western blot with quantitation of percentage of EndoH resistant LPL in (b) (n = 6 for WT, 3 for *Sel1L^AdipCre*, 4 for *Atg7^AdipCre* and *DKO*). s, i and r, EndoH sensitive, intermediate and resistant, respectively. **c, d** Representative immunofluorescent images of LPL without (c) and with co-labeling of the endothelial marker GPIHBP1 (d) in gonadal WAT (n = 3 mice per WT and DKO; n = 2 mice per *Sel1L^AdipCre* and *Atg7^AdipCre*). Capillary, intracellular and LPL CERF are marked by yellow, white, and green arrows, respectively. **e** Representative immunofluorescent images of co-labeling of LPL and cell surface marker caveolin 1 (CAV1) in gonadal WAT (n = 2 mice per genotype). Green arrows, LPL CERFs. **f** Representative EM images following immunogold labeling of LPL in gonadal WAT (n = 3 mice per genotype). Images with color coded gold particles shown on the right. Green dot line, LPL CERFs. **g** Representative immunofluorescent images of co-labeling of P62 and LPL in gonadal WAT (n = 3 mice per genotype). Red and green arrows, P62 inclusion and LPL CERFs, respectively. **h** Representative immunofluorescent images of triple labeling of LPL (gray), ubiquitin (Ub, red) and P62 (green) in gonadal WAT (n = 2 mice per genotype). Yellow and gray arrows, Ub-containing P62 inclusions and LPL CERFs, respectively. Histogram was plotted as mean with SD; each data points were derived from biologically independent mice/samples. P values were derived by one-way ANOVA followed by Tukey's test; n.s. not significant. Source data are provided as a Source Data file.

surface membrane protein caveolin1 (CAV1) (green arrows, Fig. 4e). Immunogold labeling of LPL followed by TEM imaging demonstrated that LPL was enriched in the CERFs (Fig. 4f). LPL-containing foci were distinct from the P62 inclusions in WAT as shown using confocal microscopy (Fig. 4g). Furthermore, unlike the P62 inclusions which were ubiquitin positive, LPL-containing CERFs were devoid of ubiquitin (Fig. 4h). Taken together, these data demonstrated that the LPL-containing CERFs and P62 inclusion are two distinct cellular quality-control compartments.

## LPL interacts extensively with ER chaperones in the CERFs

The identification of LPL as a protein composition of the CERFs made it possible to further explore the biophysical and biochemical nature of the CERFs. First, we performed an exploratory LPL IP-MS (without replication) in WAT to identify LPL interactomes, where two clusters of LPL-interacting proteins were found to be enriched in either WT or *DKO* WAT. Cluster 1 contained 47 protein hits with PSMs at least 2-fold higher in WT than *DKO* WAT, while Cluster 2 contained 101 hits with PSMs at least 4-fold higher in *DKO* than WT WAT (Supplementary

Fig. 6a). Cluster 1 included cytoskeleton proteins (e.g. Myosin-11 MYH11), secretory/scaffold proteins (e.g. Caveolin 1 CAV1, Clusterin CLU and Gephyrin GPHN), and ribosomal subunits (RPL3, 4, 6, 7a, 18) (Supplementary Fig. 6b and Supplementary Table 2). These Cluster 1 hits were similarly enriched in *Atg7^AdipCre^*, but not *Sel1L^AdipCre^* WAT (Supplementary Fig. 6b and Supplementary Table 2), pointing to reduced biosynthesis and trafficking of LPL in ERAD-deficient WAT. On the other hand, Cluster 2 included ER chaperones involved in post-translational modifications such as glycosylation and hydroxylation (e.g. ribophorin 1 RPN1, Malectin MLEC and procollagen-Lysine,2-oxoglutarate 5-dioxygenase 3 PLOD3), folding and degradation (e.g. HSPA5/BiP, HSP90B1/GRP94, OS9 and GPX3)[51] (Supplementary Fig. 6c and Supplementary Table 3). Most proteins in Cluster 2 were also upregulated in *Sel1L^AdipCre^*, not *Atg7^AdipCre^* WAT (Supplementary Fig. 6c and Supplementary Table 3), indicating that LPL is accumulated in the ER in the absence of ERAD. Consistently, pathway analyses of the LPL interactomes showed that, while ribosome and exosome pathways were enriched in WT WAT, glycan-related pathways and protein processing in the ER were highly represented in *DKO* WAT (Supplementary Fig. 6d).

Indeed, IP of endogenous LPL pulled down endogenous CAV1 in WT WAT, but much less so in *DKO* WAT (Fig. 5a). By contrast, endogenous LPL interacted strongly with endogenous BiP, OS9, MLEC and PLOD3 in *DKO* WAT, but not in WT WAT (Fig. 5b). Immunofluorescent staining and TEM following immunogold labeling confirmed colocalization of chaperone proteins such as OS9, MLEC and PLOD3 with LPL in the CERFs in *DKO* WAT (Fig. 5c–f and Supplementary Fig. 7a–f). Taken together, our data suggested that LPL is present, either as folding intermediates or terminally misfolded, in the CERFs of *DKO* adipocytes.

### CERFs progressively enlarge and become insoluble with age

Next, we examined the biochemical properties of CERFs using LPL as a readout. Western blot analysis following non-denaturing SDS-PAGE showed that LPL formed significantly more HMW aggregates in detergent-soluble fractions of *DKO* WAT than those in the other cohorts (Fig. 6a). These aggregates were formed via aberrant disulfide bonds as they were sensitive to the treatment of β-ME (Fig. 6a). In addition, significantly more LPL was detected in the detergent-insoluble fractions of *DKO* WAT compared to those of the other cohorts (Fig. 6b). Indeed, about half of LPL proteins were insoluble at 12 weeks of age vs. 30% at 5 weeks of age (Fig. 6c), suggesting that LPL gradually becomes insoluble with age. Similar observation was obtained for BiP (Fig. 6c). Indeed, some CERFs from 12-week-old WAT appeared hardened under TEM (Fig. 6d). In keeping with these findings, the size and abundance of LPL CERFs were markedly elevated in *DKO* WAT from 5 to 12 weeks of age (arrows, Fig. 6e and Supplementary Fig. 8a). These CERFs were not amyloidogenic as revealed by the negative Thioflavin S staining, unlike the amyloidogenic β-amyloid plaques in the brains of Alzheimer's disease 5xFAD mouse model (arrows, Supplementary Fig. 8b, c). Hence, these data demonstrated that CERFs grow and become insoluble with age.

### In vitro reconstitution of LPL condensates, coated by BiP

We noticed that, while BiP colocalized with LPL (Fig. 6e and Supplementary Fig. 8a), BiP seemed to wrap around LPL in the CERFs of *DKO* WAT in a 3D reconstruction of the confocal images (Fig. 7a, b) and immunogold co-labeling of BiP and LPL followed by TEM (Fig. 7c). This distribution pattern was not seen in other cohorts (Supplementary Fig. 9). This unique spatial organization prompted us to explore how LPL proteins assemble in CERFs by asking whether LPL protein can self-assemble or condense spontaneously in vitro using purified human recombinant LPL (Supplementary Fig. 10a, b).

Indeed, LPL spontaneously phase separated into hollow droplets in a protein concentration-dependent manner (Fig. 7d). Surprisingly,

while the addition of purified BiP had no noticeable effect on the morphology of LPL droplets, BiP formed an outer layer surrounding the LPL (Fig. 7e), bearing striking resemblance to the situation in vivo (Fig. 7a–c). Indeed, BiP coated outside of the LPL droplets, as revealed by the quantified fluorescence intensities of red and green channels (Fig. 7f), suggesting that this in vitro reconstituted structure resembles the pattern seen in CERFs. These LPL-BiP droplets were solid and spheric shaped as visualized by TEM (Fig. 7g). The turbidity of LPL droplets decreased in the presence of BiP, but not BSA (Fig. 7h). Fluorescence recovery after photobleaching (FRAP) analysis showed that LPL droplets exhibited relatively low mobility (Fig. 7i and Supplementary Fig. 10c), indicating the LPL droplets were more gel-like in nature. The addition of EGFP-BiP slightly increased LPL mobility (Fig. 7i and Supplementary Fig. 10c).

### Redox environment and C-terminal W loop influence LPL self-assembly or aggregation

We next explored the molecular mechanisms underlying LPL phase separation. LPL has five disulfide bonds (Supplementary Fig. 8a)[43]. Indeed, in the absence of a reducing agent β-mercaptoethanol (β-ME), LPL protein readily aggregates in addition to form phase-separated droplets (arrows, Fig. 8a). These aggregates could not be reversed by the addition of BiP (arrows, Fig. 8b). To further explore how LPL self-assembly occurs, we assessed the importance of LPL structural domains in this process. Truncating the C-terminal domain (CTD) of LPL significantly hampered its phase separation (Fig. 8c, d). Taking a cue from a recent study showing that the tryptophan (W) loop at the CTD of LPL is critical for the formation of helical LPL oligomers[52], we mutated three tryptophan residues (W417, W420 and W421 in human LPL) in this loop to asparagine (W3N) (Fig. 8e). Indeed, W3N mutation significantly reduced the efficiency and size of LPL droplets (Fig. 8f, h), and moreover, promoted LPL aggregation under non-reducing conditions (arrows, Fig. 8g). These data indicated that LPL self-assembly in vitro is sensitive to the redox environment and, at least partially, mediated by its C-terminal tryptophan loop.

## Discussion

Our data demonstrate a sequence of events leading to the clearance of misfolded proteins in the ER by SEL1L-HRD1 ERAD and autophagy in adipocytes in vivo (Fig. 9). The observations that SEL1L-HRD1 ERAD limits basal autophagy activity and that ER fragments containing misfolded proteins and aggregates form in *Sel1L*-deficient adipocytes establish this ERAD branch as a central player in the maintenance of ER homeostasis, at least under physiological condition. SEL1L-HRD1 ERAD not only degrades misfolded proteins, but also may affect ER-phagy activity by limiting its substrate (ER fragments) availability and the autophagy activity under basal conditions. Our data provide an in vivo visualization of ER fragments in mammalian cells, and their subsequent coalescence into CERFs when ER-phagy is absent. These findings highlight the importance of the crosstalk among the branches of ER quality control pathways in the maintenance of ER homeostasis under physiological settings, which underscores the profound ability of cells to clear misfolded proteins from the ER (Fig. 9).

Our data reveal a profound cellular adaptation capability in dealing with misfolded proteins or protein aggregates in the ER. Adipocytes were able to survive the lack of SEL1L-HRD1 ERAD, autophagy or both – two principal degradative mechanism in the cells – without instigating an overt UPR. We speculate that the formation of CERFs as a strategy to sequester potentially toxic protein aggerates in the ER, without triggering deleterious cellular consequences including overt UPR and cell death in adipocytes. CERFs are storage compartments for misfolded proteins and aggregates, which progressively enlarge and become insoluble with time. Intriguingly, this unique structure was not seen in β cells lacking SEL1L-HRD1 ERAD and autophagy as we recently reported[42]. The dynamic and intramembranous features of the CERFs

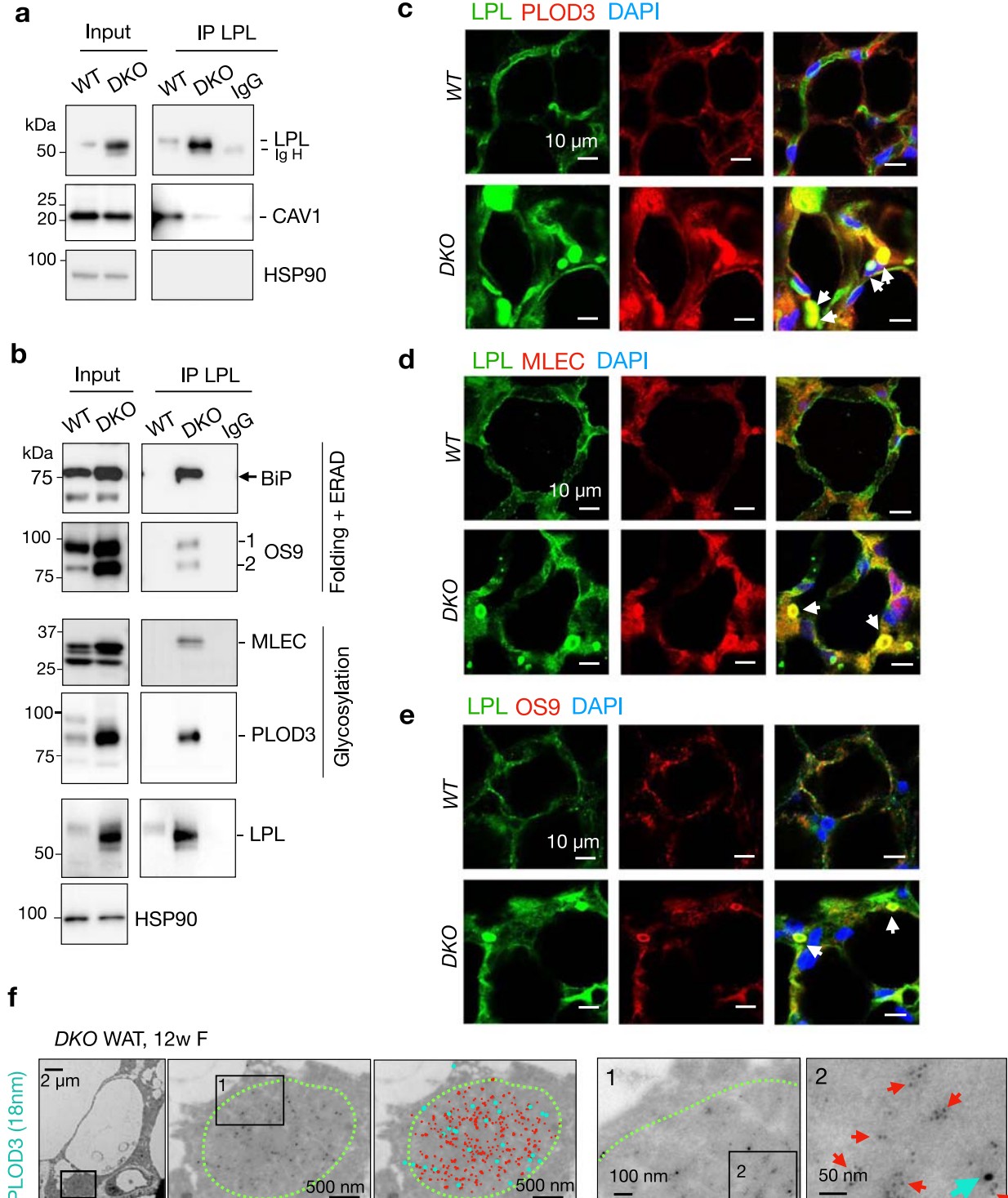

**Fig. 5 | LPL is associated with multiple ER chaperones in the CERFs.**
**a**, **b** Immunoblot analysis following immunoprecipitation of endogenous LPL in gonadal WAT (n = 3 mice each). **c**–**e** Representative immunofluorescent images of co-labeling of LPL and PLOD3 (**c**, n = 3 mice per genotype), MLEC (**d**, n = 2 mice per genotype), and OS9 (**e**, n = 2 mice per genotype), the top hits identified in cluster 2 (Supplementary Fig. 6c), in gonadal WAT. **f** Representative TEM images following immunogold labeling of LPL and PLOD3 in gonadal WAT (n = 3 mice). Green dotted line outlines the CERFs. Images with color coded gold particles shown on the right. Source data are provided as a Source Data file.

make it unique and distinct from the homogenous ER inclusions previously reported in the ER storage diseases, such as those containing fibrinogen or α1 anti-trypsin in the liver, proAVP in the AVP neurons, and CFTR disease mutant[53,54]. Using IP-MS, we identified LPL as a protein substrate of SEL1L-HRD1 ERAD and a principal component of CERFs – a significant finding as it made possible subsequent

biochemical and biophysical studies of the CERFs. Indeed, our data further show that these CREFs are also composed of many ER chaperones, which are involved in the folding and refolding of misfolded proteins or aggregates such as BiP, OS9 and MLEC.

We in vitro reconstituted the CERFs through LPL and BiP phase separation. Phase separation has been recently reported as an

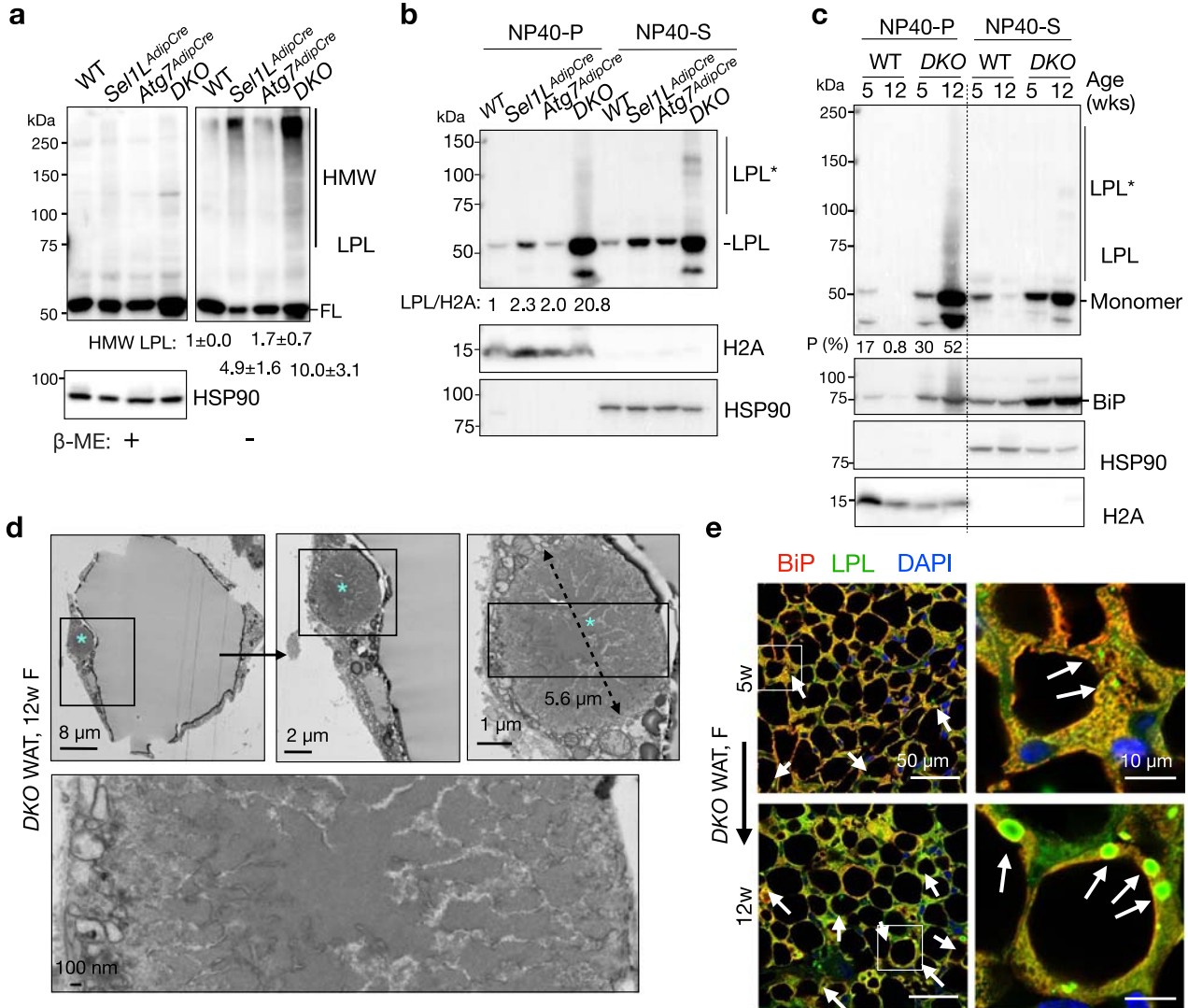

**Fig. 6 | CERFs progressively enlarge and decrease solubility with age.**
**a** Immunoblot analysis of LPL under reducing (left) and non-reducing (right) conditions in gonadal WAT lysates of 12-week-old mice (n = 4 mice per WT and DKO, and 2 mice per *Sel1L^AdipCre* and *Atg7^AdipCre*). Stats shown comparing *DKO* vs. *Sel1L^AdipCre* (P = 0.01) by one-way ANOVA followed by Tukey's test. **b** Immunoblot analysis of LPL in gonadal WAT lysates of 12-week-old mice fractionated in 1% NP40 (S, soluble; P, pellet). HSP90 and H2A, loading controls for S and P fractions, respectively (n = 3 mice per genotypes). **c** Immunoblot analysis of LPL in gonadal WAT lysates of 5- and

12-week-old WT and *DKO* mice fractionated in 1% NP40 as in b (n = 3 mice per group). LPL*: insoluble LPL that is resistant to heat and β-Mercaptoethanol (β-ME). **d** TEM images of a CERF (*) in gonadal WAT of 12-week-old mice with the diameter indicated (n = 4 mice). **e** Representative immunofluorescent images of co-labeling of LPL and BiP in gonadal WAT of mice at the age of 5 and 12 weeks, respectively (n = 3 mice per group). Arrows, LPL CERFs. Data are mean ± SD. Source data are provided as a Source Data file.

important mechanism for the formation of intracellular biological condensates[55], including the Z-α1-antitrypsin ER inclusion[56]. Consistent with the notion that LPL protein is misfolding-prone[31,43], and tends to polymerize into helical filaments[52], our data reveal that purified LPL readily forms not only precipitates, but also phase separation in vitro. Unlike many other proteins[57,58], phase separation of LPL is unlikely mediated by the low-complexity sequences as LPL is predicted to be an ordered protein by IUPred2[59]. Instead, LPL oligomerization and phase separation may require protein-protein interactions among structured domains[60]. Indeed, our data showed that both redox environment and the tryptophan loop of LPL may play important roles in its phase separation. In addition, our data showed that BiP surrounds LPL in the CERFs and droplets in vivo and in vitro, respectively, and that the interaction with BiP may alter LPL physical properties, including solubility and mobility in the CERFs. Further studies are required to delineate the importance of LPL phase separation in its basal physiological function and trafficking[61].

Our data demonstrate that SEL1L-HRD1 ERAD is a critical regulator of LPL biology by degrading misfolding LPL in the ER. If not cleared efficiently, these LPL forms aggregates in the ER, which are cleared by ER-phagy. We acknowledge that the use of *Atg7* KO mouse model is not ideal to study the ER-phagy process per se. However, despite of our limited understanding of ER-phagy in vivo, we were able to visualize the ER-phagy process specifically with the use of powerful imaging tools including confocal microscopy, immunogold labeling and TEM. One notable hit in the IP-MS of LPL is a non-canonical ER-phagy receptor CCPG1[16] – an active investigation in our laboratory.

Under basal state with functional SEL1L-HRD1 ERAD, our data show that autophagy plays a minor role in the maturation of LPL in the ER. In both WT and *Atg7^AdipCre* WAT, it should be noted that LPL does form small extra-ER puncta, likely representing those in storage vesicles as previously reported[43,52,62]. This finding is consistent with previous studies showing that inhibiting macroautophagy increased the amount of LPL at the cell surface[63]. By contrast, when SEL1L-HRD1

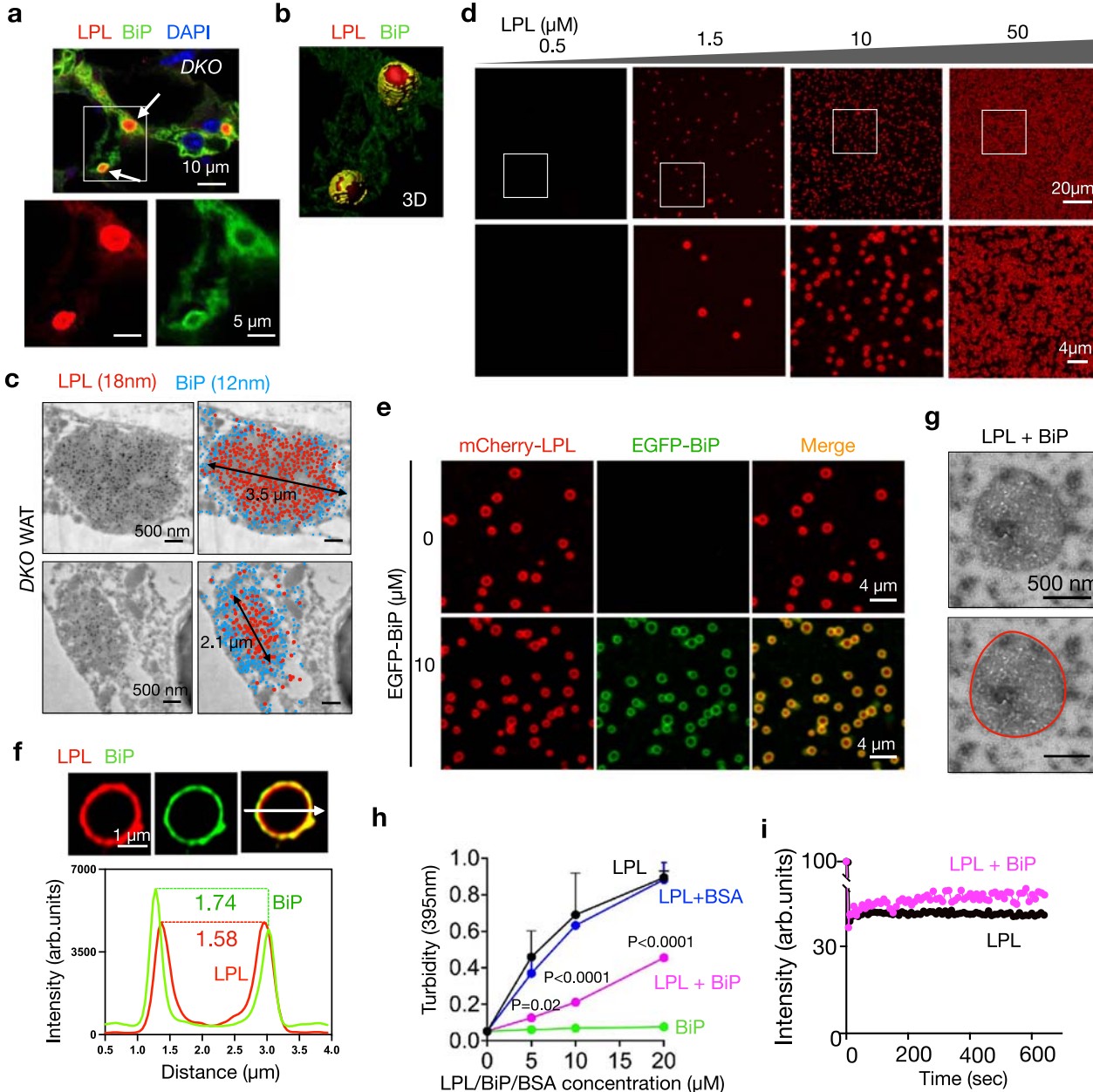

**Fig. 7 | In vitro reconstitution of LPL condensates, coated by BiP.**
**a, b** Representative immunofluorescent images of co-labeling of LPL and BiP (a) and its 3D reconstruction (b) in gonadal WAT of *DKO* mice (n = 3 mice). Arrows, CERFs. **c** Two representative EM images following immunogold co-labeling of LPL and BiP in gonadal WAT (n = 3 mice). Images with color coded gold particles against LPL (red) and BiP (blue) are shown on the right. **d** Representative images of mCherry-LPL phase-separated condensates at the indicated protein concentrations (n = 3 independent repeats). **e** Representative images of mCherry-LPL (10 μM) phase-separated condensates in the presence or absence of EGFP-BiP (n = 3 independent repeats). **f** Fluorescent images and intensity profile across the focal plane (arrow) of an LPL and BiP co-condensed droplet. **g** Negative stain TEM images showing LPL-BiP co-condensates (n = 2 independent repeats). **h** Turbidity at 395 nm (OD395) was measured for mCherry-LPL phase-separated condensates, in the presence or absence of EGFP-BiP or BSA (n = 3 independent repeats). Graph was plotted as mean with SD. P values were derived by Two-way ANOVA followed by Tukey's multiple comparisons test; n.s., not significant. **i** Fluorescence intensity recovery curves of mCherry-LPL condensates in vitro in the presence or absence of EGFP-BiP post photo-bleaching. Source data are provided as a Source Data file.

ERAD is impaired, autophagy/ER-phagy plays a major role in clearing misfolded LPL and its aggregates in fragmented ER. While molecular events underlying the formation of ER fragments and subsequent coalescence remain vague, this study uncovers the biochemical chain of events involved in disposing of misfolded protein in the ER of adipocytes. Further characterization of the pathways and mechanisms underlying the formation of ER fragments and cellular inclusions as well as their interaction with various quality control pathways in vivo is key to the understanding of the disease pathogenesis associated with ER protein misfolding, and basis for future therapy.

## Methods

### Mice

All animal procedures were approved by the Institutional Animal Care and Use Committee of the University of Michigan Medical School (PRO00010658) and in accordance with the National Institutes of Health (NIH) guidelines.

All the mouse models used in this study were on the C57BL/6 J background. *Sel1L*$^{flox/flox}$ [24], *Atg7*$^{flox/flox}$[64] and *Sel1L*$^{AdipCre}$ (*Sel1L*$^{flox/flox}$;*Adipoq-Cre*$^{+/-}$)[31] mice have been described previously. *Atg7*$^{flox/flox}$ mice were provided by Rajat Singh (University of California, Los Angeles, USA) with the

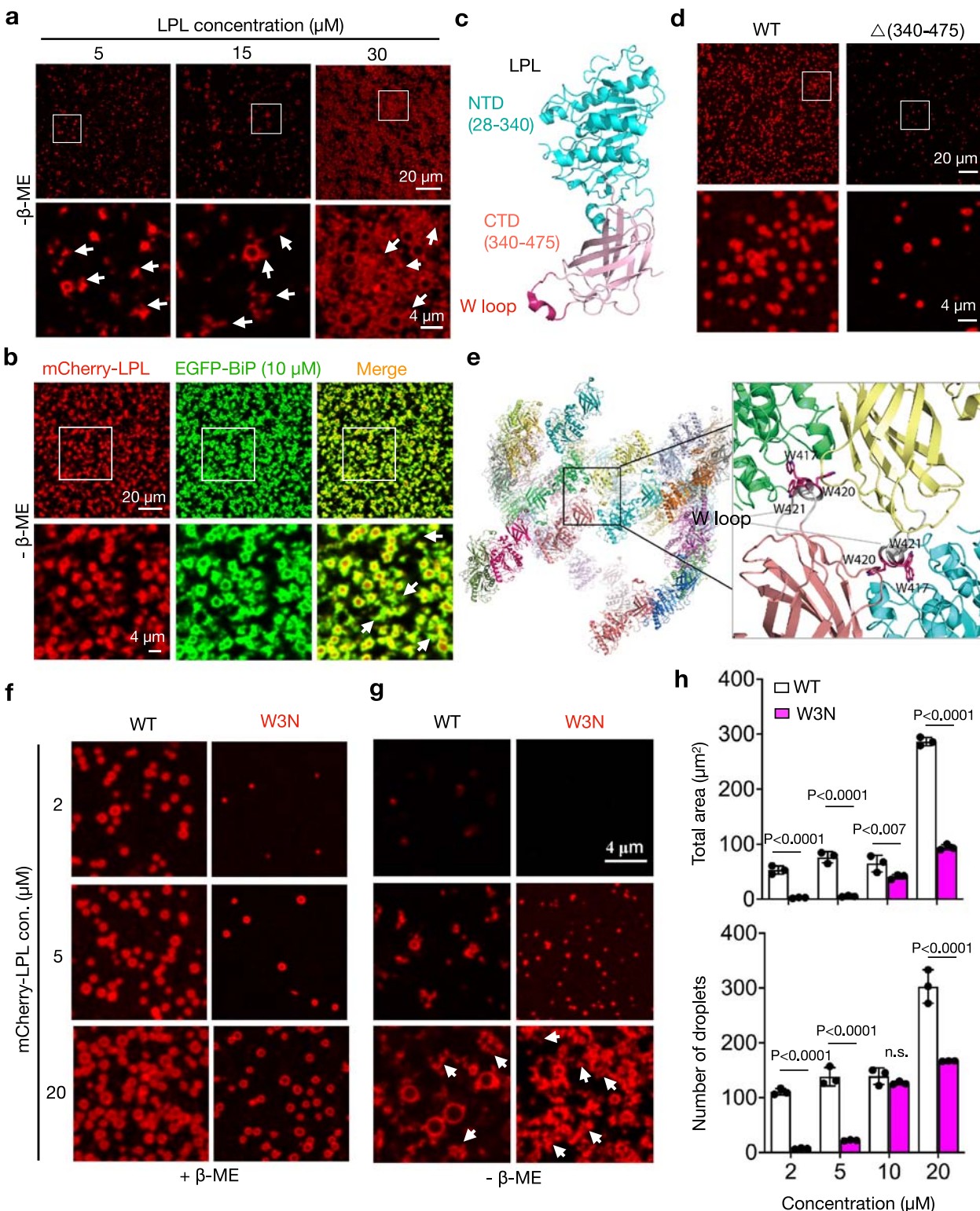

**Fig. 8 | Redox environment and C-terminal tryptophan loop influence LPL self-assembly or aggregation.** Representative images of phase-separation of mCherry-LPL (**a**) and LPL + EGFP-BiP (**b**) in the absence of β-mercaptoethanol (β-ME) (n = 3 independent repeats). Arrows, aggregates. **c** Predicted structure of LPL monomer (PDB ID: 6U7M) showing the N- (NTD, in cyan) and C-terminal domain (CTD, in pink). The tryptophan (W) loop is highlighted in magenta. **d** Representative images of phase-separated condensates of mCherry-LPL WT and the mutant lacking the CTD (amino acids 340-475) (n = 3 independent repeats). **e** Predicted structure of LPL oligomers with the inset on the right showing the tetrameric interface with two W loops colored in gray. The side chains of three tryptophan residues W417, W420, and W421 in magenta are shown as sticks. Representative images of phase-separated condensates of mCherry-LPL WT and the tryptophan mutant (W3N) in the presence (**f**) and absence (**g**) of β-ME. Arrows, aggregates (n = 3 independent repeats). **h** Quantitation of the total area and the number of droplets per unit focal plane shown in (**f**) (n = 3 independent repeats). Histogram was plotted as mean with SD; each data points were derived from biologically independent samples. P values were derived by Two-way ANOVA followed by Sidak's multiple comparison test; n.s., not significant. Source data are provided as a Source Data file.

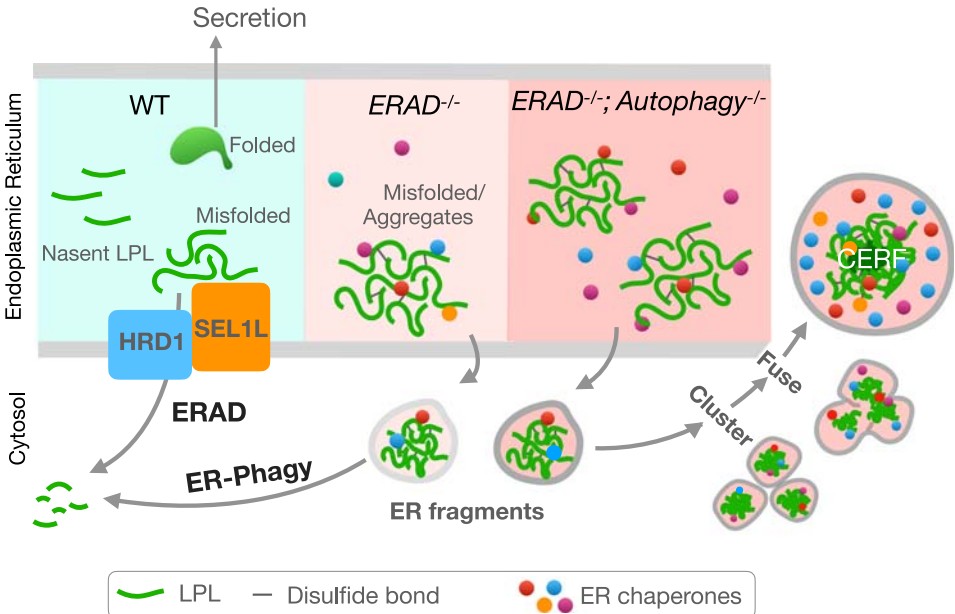

**Fig. 9 | Proposed model for the mechanism to dispose of misfolded proteins in the endoplasmic reticulum of adipocytes.** SEL1L-HRD1 ERAD plays a major role in the degradation of misfolded LPL in the ER under basal state. Without ERAD, ER fragments containing misfolded protein and aggregates form, which are cleared by autophagy, i.e. ER-phagy. In adipocytes lacking both SEL1L-HRD1 ERAD and autophagy, ER fragments accumulate, cluster and fuse to form CERFs.

permission of Masaaki Komatsu and Keiji Tanaka (Tokyo Metropolitan Institute of Medical Science, Tokyo, Japan). *Atg7^AdipCre* (*Atg7^flox/flox*;*Adipoq-Cre^+/−*) mice were generated by breeding *Atg7^flox/flox* mice with *Adipoq-Cre* mice[65]. The single knockouts were then intercrossed to generate adipocyte-specific (*Sel1L^flox/flox*;*Atg7^flox/flox*;*Adipoq-Cre^+/−*). *Sel1L^AdipCre*;*Atg7^AdipCre/+* and *Sel1L^AdipCre/+*;*Atg7^AdipCre* mice were generated as *Sel1L^AdipCre* and *Atg7^AdipCre* littermate controls, respectively. *Sel1L^flox/flox*;*Atg7^flox/flox*, *Sel1L^flox/+*;*Atg7^flox/flox*, *Sel1L^flox/flox*;*Atg7^flox/+* and *Sel1L^flox/+*;*Atg7^flox/+* littermates were used as WT controls. Brown adipocyte-specific *Sel1L^Ucp1Cre* (*Sel1L^flox/flox*;*Ucp1-Cre^+/−*) and *Hrd1^Ucp1Cre* (*Hrd1^flox/flox*;*Ucp1-Cre^+/−*) were generated by breeding *Sel1L^flox/flox* and *Hrd1^flox/flox*[66] with *Ucp1-Cre* mice (The Jackson Laboratory, #024670) as previously described in ref. 39. Experiments were performed using age- and sex-matched littermates at the ages of 5 and 10–12 weeks. The Alzheimer's disease 5xFAD female mice[67] at the age of 6 months and their WT littermates were gifts form Dr. Geoffrey Murphy. All mice were housed in a pathogen-free animal facility at 22 ± 1 °C on a 12-h light/dark cycle with 40–60% humidity and fed a low-fat diet (13% fat, 57% carbohydrate, and 30% protein, LabDiet 5LOD), unless otherwise indicated.

## Animal experiments

The fasting-refeeding and oil-gavage experiments were performed as described in ref. 31. Briefly, mice were fasted for 16 h with refeeding with HFD (60% fat, 20% carbohydrate and 20% protein, Research Diet D12492) for 3 h or olive oil gavage (1 µl/1 g body weight) for 2.5 h. For measurement of blood TG, 12-week-old mice were fasted for 16 h followed by oral gavage of olive oil. Blood was withdrawn through cardiac puncture after 2.5 h. TG levels in the serum were measured using LabAssay™ Triglyceride kit (Wako Chemicals, 290-63701) per manufacturer's instruction.

## Cell lines

Wildtype (WT) and inducible *Sel1L*-deficient (*Sel1L^ERCre*) pre-adipocytes were isolated from brown adipose tissue of mice and immortalized using SV40 large T antigen and differentiated into adipocytes as previously described in refs. 39,68. Cells were cultured at 37 °C with 5% CO₂ in DMEM with 10% fetal bovine serum (Fisher Scientific). Cultures were passaged every 2–3 days, when 80% confluence was reached. Cells were treated with 400 nM 4-hydroxytamoxifen (4-OHT, Sigma H7904) at day three of differentiation for 2 days to generate *Sel1L^−/−* adipocytes. To induce ER stress, differentiated adipocytes were incubated in fresh DMEM medium in the presence and absence of 5 µg/ml tunicamycin (Calbiochem) for 4 h as described in refs. 31,69. To measure the stability of LPL, fully differentiated adipocytes were incubated with 2 µg/ml brefeldin A (MilliporeSigma) for 30 min followed by the addition of 50 µg/ml cycloheximide (Calbiochem), as described in ref. 31.

To generate CRISPR/Cas9-based adipocytes, CRISPR-targeting oligonucleotides were designed for mouse SEL1L (5'-GCCAGCAAC-TACTTTGCCCG-3') and HRD1 (5'-ATCCATGCGGCATGTCGGGC-3'). Target sequences were inserted into lentiCRISPR v2. The third-generation lentiviral vector system (RRE [gag/pol], VSV-G and REV) were used to generate lentiviral particles in HEK293T cells. Media containing lentivirus were harvested at 24 and 48 h post-transfection for cell infection. One day after infection, pre-adipocytes positive for CRISPR/Cas9 were selected in the medium containing 2 µg/ml puromycin for 72 h and then maintained in regular medium without selection drugs.

## Western blot

Mouse tissues and cells were harvested, snap-frozen in liquid nitrogen, and processed for (non)-denaturing SDS-PAGE and Western blot as described in ref. 30. Primary antibodies used are: anti-HSP90 (1:10,000, Santa Cruz, sc-7947), anti-GAPDH (1:10,000, ProteinTech, 60004-1-Ig), anti-LPL (1:500, gift from Dr. Andre Bensadoun, Cornell University and R&D Systems AF7197-SP)[31], anti-SEL1L (1:1000, Abcam, ab78298; 1:10,000, homemade)[39], anti-HRD1 (1:200, gift from Richard Wojcikiewicz, Syracuse University; 1:1000, ProteinTech, 13473-1-AP)[31], anti-ATG7 (1:1000, Cell Signaling, 8558 S), anti-P62 (1:5000, MBL, PM066; 1:5000, Enzo, BML-PW9860), anti-LC3B (1:2000, Cell Signaling, 2775 S), anti-BiP (1:2000, Abcam ab21685), anti-OS9 (1:2000, Abcam, ab109510), anti-GLUT4 (1:2000, ProteinTech, 66846-I-Ig), anti-CALNEXIN (1:5000, ProteinTech, 10427-2-AP), anti-MLEC (1:2000, ProteinTech, 26655-1-AP), anti-PLOD3 antibody (1:2000, 11027-1-AP),

anti-CAV 1 (1:2000, ABclonal, A1555), anti-Caspase 3 (1:1000, Cell Signaling, 9662 S); anti-cleaved Caspase 3 (1:1000, Cell Signaling, 9661 S), anti-H2A (1:1000, Cell Signaling, 2578), anti-mCherry (1:1000, ABclonal, AE002), and anti-His (1:1000, ABclonal, AE003). Membranes were washed with TBST and incubated at room temperature for 1 h with secondary antibodies: donkey anti-goat IgG-HRP (1:5000, Thermo-Fisher, #PA1-28664), goat anti-rabbit IgG-HRP (1:5000, Bio-Rad, #1706515), goat anti-mouse IgG-HRP (1:5000, Bio-Rad, #1706516) and goat anti-guinea pig IgG-HRP (1:5000, Bio-Rad, #AHP863P). The uncropped blots were provided in the Source Data file.

### Detergent-solubility assay

The NP40 solubility assay was performed as described in ref. 30. Briefly, frozen WAT was lysed in cold NP40 lysis buffer (1% NP40, 50 mM Tris-HCL, PH 7.5, 150 mM NaCl, 1 mM EDTA) with complete protease inhibitors (MilliporeSigma) and PhosSTOP (MilliporeSigma) by gentle sonication. After centrifugation at 13,400 g for 20 min at 4 °C, the supernatant (NP40S fraction) was collected and protein concentration in soluble fraction was determined using Bradford assay followed by normalization. The supernatant was mixed with 5x SDS protein sample buffer and heated at 95 °C for 5 min. The pellet (NP40P fraction) was washed with lysis buffer, centrifuged to remove fat and mixed with 1x reducing SDS-sample buffer with ¼ volume of total volume of the soluble fraction. The pellet was then gently sonicated and heated at 95 °C for 30 min. Samples of soluble and pellet fractions (10 µl each) were loaded onto the gel. Bands intensity of soluble (S) and pellet (P) fractions were determined using Image lab (Bio-Rad). To quantitate the relative amount of protein in pellet fraction (P%), the band intensity of pellet fraction was divided by 4 (P/4) and P% was calculated as (P/4) / [(P/4) + S] x 100%.

### Immunoprecipitation (IP) and IP-based LC-Tandem Mass spectrometry (IP-MS)

For SEL1L IP-MS, BAT from 12-week-old female mice (one per genotype) was used. LPL IP-MS was performed in two separate experiments using frozen WAT pooled from 12-week-old mice (5 WT, 6 $Sel1L^{AdipCre}$ and 7 $DKO$ mice in experiment #1; and 4 WT and 3 $Atg7^{AdipCre}$ mice in experiment #2). Tissues were lysed in cold IP lysis buffer (1% Triton X-100, 150 mM NaCl, 25 mM Tris-HCl pH 7.5), supplemented with protease and phosphatase inhibitors. Overall 5–8 mg of protein lysates was used for IP, pre-cleared with Protein G/A agarose beads for 4 h at 4 °C. Samples were then incubated with IgG (MBL, PM094 or Cell Signaling, #2729), LPL antibody (2 mg/ml, gift from Dr. Andre Bensadoun)[31] or SEL1L antibody (2 mg/ml, homemade)[39] overnight at 4 °C. Immunocomplexes were precipitated by mixing samples with 150 µl Protein G/A agarose beads under gentle rocking for 2 h at 4°C. The beads were washed with cold IP buffer followed by cold PBS for three times at 4 °C. The beads containing immunocomplexes were stored at −80 °C until further analysis. The samples were submitted to the Proteomics Resource Facility at the University of Michigan Medical School on a fee-for-service basis. Proteins were identified by searching the MS/MS data using Proteome Discoverer (v2.4, Thermo Scientific). False discovery rate (FDR) was determined using Percolator and proteins/peptides with a FDR of ≤1% were retained for further analysis. The mass spectrometry proteomics data have been deposited to the ProteomeXchange Consortium via the PRIDE[70] partner repository with the dataset identifier PXD038310 and PXD040899. For Western blot, proteins on the beads were eluted using 2X protein sample buffer and heated at 95 °C for 5 min. For LPL Western blot, Trueblot anti-goat IgG HRP (1:1000, Rockland, mouse monoclonal eB270, #18-8814-31) was used as secondary antibody.

In SEL1L IP-MS, top protein hits enriched in $Hrd1^{Ucp1Cre}$ BAT were identified based on the criteria: peptide spectrum matches (PSM) in $Hrd1^{Ucp1Cre}$ ≥ 2, PSMs in $Hrd1^{Ucp1Cre}$ / WT ≥ 2, and PSMs in $Hrd1^{Ucp1Cre}$ /

$Sel1L^{Ucp1Cre}$ ≥ 4 and PSMs in $Hrd1^{Ucp1Cre}$ / IgG ≥ 4. The full list of identified hits with raw PSM values was provided in Supplementary Table 1.

In LPL IP-MS: protein hits enriched in WT versus $DKO$ WAT (Cluster 1) were identified based on the criteria: PSMs in WT/ $DKO$ ≥ 2, PSMs in WT ≥ 2 and PSMs in IgG =0. Protein hits enriched in $DKO$ versus WT (Cluster 2) were identified based on the criteria: PSMs in $DKO$ / WT ≥ 4, PSMs in $DKO$ ≥ 4, and PSMs in IgG =0. The full list of identified hits with raw PSM values was provided in Supplementary Tables 2 and 3. Heatmaps shown in Supplementary Fig. 6b and c were presented as fold change to WT in each experiment.

### Histology and immunofluorescent staining

WAT was dissected and fixed in 10% neutral buffered formalin (VWR 95042-908) and further processed for embedding, sectioning and H&E staining by the Histology Core at the University of Michigan Medical School. The H&E slides were imaged using Aperio Scanscope (Leica). TUNEL labeling was performed per manufacturer's protocol using the In-Situ Cell Death detection kit (Roche, 11684795910). For positive control, paraffin section of WAT was incubated with 1500 U/ml recombinant DNase I (Omega, E1091) at 37 °C for 30 min per manufacture's instruction.

For immunofluorescence labeling, paraffin-embedded WAT was sectioned at 8 µm and deparaffinized in xylene and re-hydrated using graded ethanol series and distilled water. The antigen retrieval was performed by boiling the slides in 1 mM EDTA (pH 8.0) or citric acid buffer (pH 6.0) using microwave oven for 25 min, followed by blocking (10% normal donkey serum, 0.4% tritonX-100 in PBS) at room temperature for 1 h. The slides were incubated at 4 °C overnight with primary antibody: anti-LPL (1:200, gift from Dr. Andre Bensadoun;[31] 1:100, R&D Systems AF7197-SP), anti-GPIHBP1 (1:250, gift from Dr. Stephen Young);[71] anti-KDEL (1:500, Novus NBP1-97469), anti-BiP (1:500, Abcam ab21685), anti-P62 (1:500, MBL, PM066), anti-P62 (1:500, Enzo, BML-PW9860), anti-OS9 (1:250, Abcam, ab109510), anti-PLOD3 antibody (1:500, ProteinTech,11027-1-AP), anti-CAV1 (1:250, ABclonal, A1555), anti-MLEC (1:200, ProteinTech, 26655-1-AP). Next day, the slides were incubated at room temperature for 1 h with secondary antibodies: 1:500 Alexa fluor 488 affinipure donkey anti-goat IgG (H + L) (Jackson ImmunoResearch, 705-546-147); 1:500 Alexa fluor plus 555 donkey anti-goat IgG (H + L) (Invitrogen, A32816); 1:500 Alexa fluor plus 647 donkey anti-goat IgG (H + L) (Invitrogen, A32849); 1:500 Alexa fluor 647 affinipure donkey anti-rat IgG (H + L) (Jackson ImmunoResearch, 712-606-150); 1:500 Alexa fluor plus 555 donkey anti-mouse IgG (H + L) (Invitrogen, A32773); 1:500 Alexa fluor 488 affinipure donkey anti-rabbit IgG (H + L) (Jackson ImmunoResearch, 711-546-152); 1:500 Alexa fluor plus 555 donkey anti-rabbit IgG (H + L) (Invitrogen, A32794); 1:500 Alexa fluor 647 affinipure donkey anti-rabbit IgG (H + L) (Jackson ImmunoResearch, 711-606-152); and 1:500 Alexa fluor 647 affinipure donkey anti-guinea pig IgG (H + L) (Jackson ImmunoResearch, 706-606-148). Both primary and secondary antibodies were diluted in PBS containing 10% normal donkey serum and 0.2% triton X-100. The nuclei and slides were stained and mounted by prolong gold antifade reagent with DAPI (ThermoFisher) followed by imaging using the Nikon A1 confocal microscope in Microscopy and Imaging Analysis Core (Michigan Diabetes Research Center, University of Michigan Medical School). The 3D re-construction of the immunofluorescent image was performed by Imaris (Oxford Instruments) at the Microscopy and Imaging Analysis Core.

### Thioflavin S staining

For thioflavin staining, the paraffin-embedded WAT and brain sections were deparaffinized as mentioned above. The slides were then incubated with 1% thioflavin for 1 h at room temperature in dark followed by a series of washing steps in 80% ethanol, 70% ethanol, 50% ethanol and distilled water. Then, the slides were mounted and dried overnight

followed by imaging using the Leica SP5 confocal microscope at the Microscopy and Imaging Analysis Core.

## Transmission electron microscopy (TEM)

The TEM of adipose tissue was performed as described in ref. 39. Briefly, mice were anesthetized and perfused with 2.5% glutaraldehyde, 4% formaldehyde in 0.1 M sodium cacodylate buffer (Electron Microscopy Sciences). WAT and BAT were immediately dissected and fixed overnight at 4 °C in the perfusion buffer. In some experiments, WAT was quickly removed without perfusion and fully immersed in the fixation buffer (2.5% glutaraldehyde and 4% paraformaldehyde in 0.1 M sodium cacodylate) for 1-h incubation at room temperature and overnight incubation at 4 °C. The fixed tissue was cut into small pieces (about 1-2 mm cubes) and then submitted to the University of Michigan Microscopy Core for washing, embedding and sectioning at 70 nm. The grid containing tissue sections was imaged by a JEOL 1400-plus electron microscope (JEOL) at the University of Michigan Microscopy Core.

## Focused ion beam-SEM (FIB-SEM)

The FIB-SEM was performed as described previously in ref. 39. In brief, the fixed white adipose tissue cubes were sent to the University of Michigan Microscopy Core for embedding. The sample block was then submitted at the Michigan Center for Materials Characterization (University of Michigan) for FIB-SEM analysis using a FEI Helios Nanolab 650 DualBeam. After determining the location of adipocyte with CERFs, the block was sequentially cut at 10 nm and imaged to obtain an image stack for tomographic reconstruction. The field of view of the SEM and the slice thickness were set to produce voxel size $5 \times 6 \times 10$. The resulting stack of images was aligned and reconstructed using Avizo v.9.3 (Thermo Fisher Scientific) and Imaris x64 9.5.1 (Oxford Instruments). Segmentation was performed using AIVIA 10 (Leica): manual segmentation was performed to determine the boundaries of the CERF in 30 images of the tomographic dataset, which was then used to train the software to segment remaining images.

## Immunogold labeling for TEM

WAT was fixed in buffer containing 3% formaldehyde, 0.2 M sucrose in 0.1 M Sorensen's phosphate buffer (PH = 7.2) and cut into small pieces and processed by the University of Michigan Microscopy Core. The ultra-thin sections (80 nm) were cut, placed bare nickel grid and stored at 4 °C. For immunogold labeling, the grids were quenched in 80 mM glycine and incubated in blocking solution (EMS 25599) containing 0.2% Tween-20 for 1 h at room temperature followed by overnight incubation with primary antibodies: anti-LPL (1:70, gift from Dr. Andre Bensadoun; 1:50, R&D Systems AF7197-SP), anti-KDEL (1:50, Novus NBP1-97469), anti-BiP (1:50, Abcam ab21685), PLOD3 (1:10, ProteinTech,11027-1-AP), and anti-P62 (1:50, Enzo, BML-PW9860) at 4 °C in a humidifying chamber. The grids were next incubated with colloidal gold antibodies (1:25, Jackson ImmunoResearch) for 1 h at room temperature followed by post-fixation in 1% glutaraldehyde and contrast staining using 0.5% uranyl acetate. After carbon evaporation, the grids were imaged by JEOL electron microscope at the University of Michigan Microscopy Core.

## RT-PCR, qPCR and Microarray

Total RNA was extracted from tissues and cells using Trizol and BCP phase separation reagent (Molecular Research Center). RT−PCR for *Xbp1* splicing (F:ACGAGGTTCCAGAGGTGGAG; R: AAGAGGCAACA GTGTCAGAG) and *l32* (F: GAGCAACAAGAAAACCAAGCA; R: TGCACAC AAGCCATCTACTCA) was performed as previously described in ref. 69. The ratio of *Xbp1s* level to total *Xbp1* level was quantified by Image lab software (BioRad). All qPCR data for mouse tissues were normalized to the expression level of ribosomal *18 s*. qPCR primer sequences are: *Lpl*

F: CGAGAGGATCCGAGTGAAAG, R: TTTGTCCAGTGTCAGCCAGA; and *18 s* F: GCAATTATTCCCCATGAACG, R: GGGACTTAATCAACGCAAGC. Microarray data[31] (accession number GSE56918) were processed using the limma package[72] (version 3.52.0) in the R environment (version 4.2.0). Genes with higher mean intensities in *Sel1L^AdipCre* than in WT samples and q values smaller than 0.05 from 2-tailed paired intensity-based moderated t-statistics (IBMT) were selected as input for pathway enrichment analysis. The pathways enrichment analysis was performed using the enrich KEGG function implemented in the cluster-Profiler package[73] (version 4.4.1) in R, with the background genes adjusted based on the genes detected in the microarray data.

## Protein expression and purification

Human LPL (amino acids 1−475) or BiP (amino acids 1-645) was inserted into pETL7a vector, with mCherry or EGFP tag at the N-terminus and 8×His-tag at the C-terminus. LPL truncations or mutations were constructed using standard mutagenesis procedures. EGFP-BiP, mCherry-LPL, and their mutants were over-expressed in *E. coli* strain BL21 (DE3). The bacteria were cultured in LB medium at 37 °C in a shaker incubator to OD600 0.6-0.8, then induced with 0.5 mM IPTG for 16 h at 16 °C. The bacteria were collected, resuspended in lysis buffer (20 mM Tris-HCl, 150 mM NaCl, 20 mM β-ME, 4 M Urea, 5% glycerol, 0.1 mM PMSF, 1× protease inhibitor cocktail, pH 7.5), and sonicated for 30 min on ice (180 W, 10 s on and 10 s off). The lysates were collected, and the supernatant was loaded onto Ni$^{2+}$-NTA resin. The column was washed with wash buffer (lysis buffer with 30 mM imidazole) and eluted with elution buffer (lysis buffer with 300 mM imidazole). Protein purity was confirmed by SDS-PAGE.

## In vitro phase separation

To obtain phase-separated mCherry-LPL and EGFP-BiP condensates, purified recombinant proteins were dialyzed at 4 °C overnight in phase separation buffer (20 mM Tris-HCl, 150 mM NaCl, 20 mM β-ME, 1 M Urea, pH 7.5). Images of the condensates were taken by Olympus SpinSR spinning disk confocal super-resolution microscope. For negative staining, the phase separated droplets were stained with 1 % uranyl acetate and imaged by Hitachi HT7800 transmission electron microscope at Tsinghua University.

## Fluorescence recovery after photobleaching (FRAP)

FRAP experiments were carried out using Nikon A1 confocal microscope. The mCherry-LPL with or without EGFP-BiP were bleached using 488-nm or 561-nm laser beam. Recovery from photobleaching was recorded at the indicated times.

## Turbidity assays

Proteins were diluted in a buffer containing 20 mM Tris-HCl, 150 mM NaCl, and pH 7.5, to desired concentrations. The absorbance at 395 nm (turbidity) was monitored by the Bio-Tek Cytation 5 imaging reader.

## Statistical analyses

All results are presented as mean ± standard deviation (SD) using Graphpad Prism8 (Graph Pad Software Inc.). Most experiments have been independently repeated for at least twice with multiple independent biological samples from which representative data are shown. IP-MS were performed once as the experiments were intended to identify hits and generate hypothesis. Statistical differences were performed using unpaired two-tailed Student's *t*-test for two groups and One-way or Two-way ANOVA analysis with a post hoc Tukey's multiple comparisons test for more than 2 groups. P values less than 0.05 were considered statistically significant. The frequency distribution of lipid droplet size between WT and *DKO* WAT was compared using Chi-square test. In Supplementary Fig.6d, the P values are calculated by two-tailed rate ratio tests and adjusted for multiple testing

by False Discovery Rate (FDR). The details for adjustment were described previously in ref. 74.

## Reporting summary

Further information on research design is available in the Nature Portfolio Reporting Summary linked to this article.

## Data availability

Proteomics datasets have been deposited to the ProteomeXchange Consortium via the PRIDE partner repository with the dataset identifiers PXD038310 and PXD040899. The predicted structure of LPL is available at PDB ID: 6U7B. Other data supporting the findings of this study are available within the article and in the Supplementary Information and Source Data files. Source data are provided with the paper. Source data are provided with this paper.

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

## Acknowledgements

We thank Drs. Andre Bensadoun, Geoffrey Murphy, and Stephen Young for providing reagents; Dr. Haibo Sha at Novartis Institutes for BioMedical Research for some pilot experiments; Dr Saskia Neher and members in the Arvan/Qi laboratories for technical assistance and insightful discussions. We also acknowledge the use of the instruments and the staff assistance from the Proteomics Resource Facility, the Microscopy Core, Microscopy and Imaging Analysis Core at the Uni-versity of Michigan. This work is supported by grants from the Ministry of Science and Technology (2022ZD0213900 and 2022ZD0204900), National Natural Science Foundation of China (32170684 to Y.L), R01DK128077, R01DK132068 (S.S.), 1R01DK120330 (L.Q. and D.F.), 1R01DK120047 (L.Q.), and Protein Folding Disease Initiative (L.Q. and M.I.). S.A.W. is supported by American Heart Association Pre-doctoral Fellowship (828841), and Barbour Scholarship and Rackham Interna-tional Student Fellowship at University of Michigan. X.W. is supported in part by Pandemic Research Recovery Grant (U078128). L.L.L. is sup-ported in part by National Ataxia Foundation Post-Doctoral Fellowship (918037). M.T. is a Pew Latin American Postdoctoral Fellow.

## Author contributions

S.A.W. designed and performed most experiments; Y.L. and C.S. designed and performed the in vitro experiments; X.W. and Y.L. per-formed analysis of proteomics and microarray data; X.Z., M.T., S.W., X.C., L.L.L., A.H.H., and H.W. assisted with some experiments; S.S., D.F., and M.I. provided reagents and insightful discussion; L.Q. designed and directed the study; L.Q. and S.A.W. wrote the manuscript; all authors commented on and approved the manuscript.

## Competing interests

The authors declare no competing interests.
