## [Peer review file · Nature Communications]

REVIEWER COMMENTS

Reviewer #1 (Remarks to the Author):

In this manuscript Wu et al. report on a novel substrate of SEL1- dependent ERAD - lipoprotein lipase (LPL) – and how the cell is dealing with this error prone protein once ERAD is deficient. In this context, the authors analyse the interplay between ERAD and the autophagic pathway and discover compensatory mechanisms of the two main degradative pathways. They also identify in cells and reconstitute in vitro a mechanism by which LPL is able to coalesce upon accumulation, which is presumably important for its degradation.

The presented work is of high quality, of considerable interest for different fields dealing with protein quality control (ERAD, autophagy, cell homeostasis), and therefore in principle very suitable for publication in Nat. Com. However, some interpretations may need to be reviewed and some additional key experiments would make the presented work much more comprehensive and impactful.

Major points:

1) The authors describe phase-separated LPL as a novel quality-control compartment, which they term CERF.

In my opinion this interpretation is misleading. A quality control compartment is a place where active quality control takes place. The identified body however seems to rather be a storage compartment, which is presumably present under basal conditions (to be confirmed) and enlarges upon pathway deficiency leading to the accumulation of LPL. I would therefore strongly suggest to overthink this interpretation.

Of note – I believe that the finding is extremely interesting and follows a predicted, yet undiscovered mechanism for ER-phagy substrate recruitment/accumulation. Interpreted in this way, the findings of Wu et al would establish phase separation within the ER as an existing, physiological highly relevant substrate recruitment process and in addition validate BiP as a co-receptor (potentially for CCPG1) within ER-phagy.

2) The authors claim to focus on the interplay between ERAD and ERphagy.

As such the LPL IP-MS experiments presented in Fig 4 should be also done in both single KO cells.

Small distinct puncta of LPL are also visible ATG7 KO cells (should be mentioned/discussed). Also the presence of CCPG1 in cluster 2 should be mentioned earlier in the result part / discussed in more length as this is a major ER-phagy player identified, known to be specifically important for certain ER stress conditions. Together this implicates that LPL is also degraded (to a small amount) by ER-phagy while ERAD is functional, presumably via the presented mechanism of coalesced, phase-separated. Parts of Cluster 2 would presumably show up under ATG7 KO conditions and help to interpret findings.

Perform co-stainings with CCPG1 and LPL | ATG7 and DKO cells.

In this context, please indicate in figures if the differences between WT and ATG7KO samples are already significant or still n.s. for Fig. 3b (LPL) and ext Fig 2c (here also use a presentation style for y-axis as in ext Fig 1a).

Fig 3b – a better neg control should be found. Even though the quantifications indicate n.s., the chosen blot clearly shows an impact of SEL1-loss on GLUT4. Plus blot for CCPG1.

3) Might have missed it – are found KDEL clusters in DKO surrounded by a single membrane (one big phase separated body within the ER/ER fragment) and do several of those accumulate? Please revise description of findings shown in Fig 2. It is also unclear, if you see extended ER-tubules and CERFs.

4) Single channels of Fig. 3c, 4g, and 5f should be shown (in main or sup figures). Here a grey scale presentation instead of colored mode should be used to avoid bias. Other single channel presentations of the figures should be also presented in/changed to grey scale.

Minor points:

- Line 62: replace unknown (not true as there are some publications related to this topic) with poorly understood.

- Line 302-303: The authors did not show that SEL1L-HRD1 ERAD regulates ER-phagy. Please remove/adapt this sentence.

Reviewer #2 (Remarks to the Author):

The physiological roles of ER-phagy and ERAD, and the intersection between these, is poorly understood. Wu et al provide data using transgenic ERAD-deficient or macroautophagy-deficient mice that the two processes compensate one another in ER proteostasis. In particular, preventing the formation of ER chaperone-delimited LDL protein clusters, composed of multiple fragments of ER, in adipose tissue.

The experimental data demonstrating this phenomenon are clear, robust and will be of great interest to the field. I recommend eventual publication.

My only caveats are in respect of the framing of the findings. These do not necessarily require additional experimentation to address:

1) The CERF compartment wherein the LDL aggregates reside could potentially be another layer of quality control (it for certain indicates a redundancy between autophagy and ERAD in quality control). However, in the absence of a genetic intervention that disrupts formation of this compartment, the authors should not imply a definitive cytoprotective effect of CERF formation.

2) Please clarify: the CERF-unrelated p62 compartments that form in DKO cells. Are these seen in ATG7 deficient tissue or is this just in DKO? If just in DKO, what are these and why do they form in response to combined ERAD and ER-phagy inhibition. Are they a parallel pathway to CERFs?

3) Technically, the authors inhibit global macroautophagy not just ER-phagy upon ATG7 knockout. Although I agree the basis of the CERF formation is likely to be defective ER-phagy, the authors should be clear on this limitation of their study within their writing (perhaps in future mouse strain studies ER-phagy could be targeted directly).

In respect of point 3, I note that CCPG1 ER-phagy receptor is detected in the mass spec. Labelling of the CERFs with this or other ER-phagy proteins (FAM134B/C?) could provide some further evidence of the role of ER-phagy. However, I am aware of the severe technical limitations (maybe impossibility) of tissue-level detection of CCPG1 protein, in particular, so don't think it is fair to require this for publication.

Reviewer #3 (Remarks to the Author):

The authors have used knockout mice for adipocyte specific gene knockouts for white and brown fat. Sel1L deficient adipocytes were also generated and treated or not with tunicamycin, brefeldin A and cycloheximide. Adipocytes with CRISPR/Cas9 targeted Sel1L and HRD1 from lentivirus transduction were also generated. The above were used to test for compensatory mechanisms in response to impaired SEL1L-HRD1 ERAD.

The authors report Xbp1 splicing in wt and SEL1L AdipCre white adipose tissue , cleaved caspase 3 and TUNEL staining, cDNA profiling microarray analysis, LC3-I, LC3-II , P62, and TEM imaging. They report for double KO Sel1L/Atg7, western blots of SEL1L, HRD1, ATG7 in white adipose tissue, size and weight of white fat pads, histology, caspase 3 cleavage and Tunel staining and Xbp1 splicing , immunogold labeling of P62 ,KDEL, and TEM of white and brown adipose tissue.

The authors report on proteins characterized in immune-precipitates of SEL1L from brown adipose tissue deleted for HRD1. Lipoprotein lipase was selected for further study with western blots and qPCR reported for HRD1 deleted adipocytes. Immunofluorescence and immunogold labeling and TEM of white adipose tissue of mice deleted for Sel1L and double KO Sel1L/Atg7 is reported as well as serum triglyceride. The authors report as well, proteins characterized in immune-precipitates of lipoprotein lipase from double KO white adipose tissue by mass spectrometry, western blots, immune-fluorescence. The authors also report in vitro phase separation of mCherry lipoprotein lipase wt or mutant, and EGFP-BiP after overnight dialysis in buffer and confocal microscopy, immunofluorescence, FRAP, or immunogold labeling and TEM.

The authors conclude that a new quality control compartment (CERF) sequesters protein aggregates in the ER.

The authors' conclusion may need a consideration of each of the individual experiments in this manuscript as well as papers published from others over the years concerning quality control compartments.

For the individual experiments, are the observations in extended data Figures 1a,b,c,d taken from experiments done for ref.28? Have the experiments in Fig1 been done on n=3 biological replicates ? For example does n=3 for TEM represent different animals? Why is the magnification for WT in Fig 1b lower than Sel1L adipcre and not indicated for wt . The reported double membranes (mitochondria of course have double membranes normally) or ER like structures are not obvious in Fig1b or extended data Fig 1e. For Fig 1c how was the quantification done especially since the cells are large and the data are expressed per adipocyte ? This is not indicated in the Experimental Procedures. For all Figures with histograms showing means and SEM, are they with biological replicates and could they be indicated with

SD that more accurately shows variations in the observations? For Figs 2b and extended data Fig 3b again what is claimed in the text is not seen in the images – where are the inner and outer membranes and are they membranes? A minor point is that the mag bars are incorrect at least for one of images in extended data Fig3b. For the IP MS experiments, the reporting is incomplete and unclear. The authors may have used peptide-spectrum matches but this is not defined. Missing are n=3 biological replicates for the IP-MS experiments, the peak lists, search engine, sequence database, enzymes used, false discovery rates at peptide and protein levels as well as what the numbers mean inside the extended data tables 1,2,3. Are these all the proteins characterized in the IP-MS? There are several other additional needed criteria for reporting the authors may consider for mass-spec based data. How were the data acquired for Fig3a, what is a hit? What are the clusters in Fig 4a,b,c,d? A minor point is for the authors to check if they confused possibly the interpretation of the glucuronosyl transferase protein they likely characterized with possibly the calnexin cycle constituent glucosyl transferase that was not? For Fig3d, Fig4h the images are difficult to see as are those in Fig5d, and extended data Fig7a. Each figure and inset in the Figures and extended data figures may be considered by the authors to assess if they really show what is claimed. Finally for the quality control compartment, have the authors considered relevance to the publications of the Kopito, Lederkremer, Klumperman, Lippincott-Schwarz, Ferro-Novick labs?

We thank all three reviewers for their insightful and constructive comments – very helpful indeed! We now have carefully addressed all the comments from the reviewers, which have been instrumental to further improve and strengthen our manuscript.

Reviewer #1:

Remarks to the Author:

In this manuscript Wu et al. report on a novel substrate of SEL1- dependent ERAD - lipoprotein lipase (LPL) – and how the cell is dealing with this error prone protein once ERAD is deficient. In this context, the authors analyse the interplay between ERAD and the autophagic pathway and discover compensatory mechanisms of the two main degradative pathways. They also identify in cells and reconstitute in vitro a mechanism by which LPL is able to coalesce upon accumulation, which is presumably important for its degradation.

The presented work is of high quality, of considerable interest for different fields dealing with protein quality control (ERAD, autophagy, cell homeostasis), and therefore in principle very suitable for publication in Nat. Com. However, some interpretations may need to be reviewed and some additional key experiments would make the presented work much more comprehensive and impactful.

We thank the reviewer for his/her positive and insightful comments on our work.

Major points

1) The authors describe phase-separated LPL as a novel quality-control compartment, which they term CERF. In my opinion this interpretation is misleading. A quality control compartment is a place where active quality control takes place. The identified body however seems to rather be a storage compartment, which is presumably present under basal conditions (to be confirmed) and enlarges upon pathway deficiency leading to the accumulation of LPL. I would therefore strongly suggest to overthink this interpretation. Of note – I believe that the finding is extremely interesting and follows a predicted, yet undiscovered mechanism for ER-phagy substrate recruitment/accumulation. Interpreted in this way, the findings of Wu et al would establish phase separation within the ER as an existing, physiological highly relevant substrate recruitment process and in addition validate BiP as a co-receptor (potentially for CCPG1) within ER-phagy. We thank and agree with the reviewer on this great point. We currently have no evidence of an active quality control. Hence, we have made changes accordingly in the title and throughout the text to reflect the nature of CERF being a storage compartment when ERAD and autophagy are compromised. We also have added a discussion on this point on page 10 and pasted below. Specifically, we now have changed the title to “The mechanism to dispose of misfolded proteins in the endoplasmic reticulum of adipocytes” from “Coalescence of endoplasmic reticulum fragments form a new quality-control compartment”.

“Our data reveal a profound cellular adaptation capability in dealing with misfolded proteins or protein aggregates in the ER. Adipocytes were able to survive the lack of both SEL1L-HRD1 ERAD and autophagy – two principal degradative mechanism in the cells – with remarkably mild ER stress. Although lacking substantial direct evidence, we propose that the formation of CERFs as a novel strategy to sequester potentially toxic protein aggerates in the ER, without triggering deleterious cellular consequences including overt UPR and cell death in adipocytes. CERFs are storage compartments for misfolded proteins and aggregates, which progressively enlarge and become insoluble with time. Intriguingly, this unique structure was not seen in β cells lacking SEL1L-HRD1 ERAD and autophagy as we recently reported.”

2) The authors claim to focus on the interplay between ERAD and ERphagy. As such the LPL IP-MS experiments presented in Fig 4 should be also done in both single KO cells. Small distinct puncta of LPL are also visible ATG7 KO cells (should be mentioned/discussed). Also the presence of CCPG1 in cluster 2 should be mentioned earlier in the result part / discussed in more length as this is a major ER-phagy player identified, known to be specifically important for certain ER stress conditions. Together this implicates that LPL is also degraded (to a small amount) by ER-phagy while ERAD is functional, presumably via the presented mechanism of coalesced, phase-separated. Parts of Cluster 2 would presumably show up under ATG7 KO conditions and help to interpret findings. Perform co-stainings with CCPG1 and LPL in ATG7 and DKO cells. In this context, please indicate in figures if the differences between WT and ATG7KO samples

are already significant or still n.s. for Fig. 3b (LPL) and ext Fig 2c (here also use a presentation style for y-axis as in ext Fig 1a).

We thank the reviewer for this great comment. First, we are exploring the role of CCPG1 in this process – excitingly, we found that CCPG1 may link ERAD to ER-phagy (Response Fig. 1): (a) validation of a good CCPG1 Ab for western blot; (b) increased protein levels of CCPG1 in *DKO* WAT and modest increase in *Atg7^{AdipCre}* compared to WT WAT; (c) interaction between LPL and CCPG1 in *DKO* WAT; (d) stabilization of CCPG1 protein in SEL1L-deficient cells. Given these interesting findings, we feel that CCPG1 and its

interaction with ERAD in LPL degradation itself is an independent project and is clearly beyond the scope of this study. We also mentioned this in the revised manuscript on page 11 and pasted below.

“One notable hit in the IP-MS of LPL is the non-canonical ER-phagy receptor CCPG1– an active investigation in the laboratory.”

Response Figure 1. CCPG1 may link ERAD to ER-phagy.

(a) Validation of mouse and human CCPG1 Western blot using the Proteintech antibody 13861-1-AP. CCPG1^{CRISPR}; CCPG1 KO cells by CRISPR CAS9; *DKO*, *Sel1L* and *Atg7* double KO white adipose tissue (WAT). (b) Immunoblot analysis of CCPG1 in gonadal WAT. (c) LPL IP in gonadal WAT using the Proteintech antibody 13861-1-AP. (d) Immunoblot analysis of endogenous ER-phagy receptors in HEK 293T cells following cycloheximide (CHX) treatment for the indicated times. CCPG1 Ab (Cell Signaling Technology antibody #80158).

Second, we have performed LPL IP-MS in gWAT of the two single KO mice. In line with our results that LPL was accumulated in the ER in the absence of ERAD (Sha et al., 2014 and this study). LPL interacted strongly with ER chaperones such as OS9, PLOD3 and MLEC in *Sel1L^{AdipCre}* and *DKO* WAT compared to those in *Atg7^{AdipCre}* and WT WAT (Response Fig. 2b and Supplementary table 3). By contrast, LPL in *Atg7^{AdipCre}* WAT pulled down hits that were similar to those of WT WAT, including secretory proteins (e.g.

CAV1) and ribosomal subunits (Response Fig. 2a and Supplementary table 2). Hence, in the absence of SEL1L ERAD, LPL is trapped intracellularly in the ER, but not in adipocytes lacking autophagy. Of note, CCPG1 strongly interacted with LPL only in *DKO* WAT, but not in the other three genotypes (Response Fig. 2b-c and Supplementary table 3).

Response Figure 2. LPL interacts extensively with ER chaperones in *Sel1L^{AdipCre}*, but not *Atg7^{AdipCre}*, WAT.

(a-b) Top-ranked protein hits that were up- (a) and down-regulated (b) proteins in *DKO* versus WT WAT by LPL IP. These data were shown in Fig.5b-c of the revised manuscript. (c) Endogenous LPL IP in gonadal WAT.

Third, our EndoH digestion assay showed that *Atg7* deficiency does not affect the ratio of EndoH resistant vs. sensitive LPL compared to that in WT cells (Response Fig. 3a), suggesting that autophagy deficiency does not affect the maturation of nascent LPL in the ER. Moreover, we observed small LPL-containing puncta in both WT and *Atg7^{AdipCre}* adipocytes (arrows, Response Fig. 3b-c), but these LPL-containing puncta in *Atg7^{AdipCre}* adipocytes do not colocalize with the ER marker KDEL (arrows, Response Fig. 3c). Hence, these findings led to our conclusion that LPL puncta formed in *Atg7^{AdipCre}* adipocytes represent post-ER trafficking vesicles or LPL storage vesicles as previously reported¹⁻³, rather than misfolded ones in the ER. Our conclusion is further corroborated by previous studies showing that inhibition of autophagy increases the amount of LPL at the cell surface⁴ and that LPL proteins form helical oligomers in storage vesicles^{1,2}.

Response Figure 3. It is unlikely that ER-phagy plays a major role in LPL maturation in the ER under basal conditions when ERAD is functional. (a) EndoH analysis of LPL in gonadal WAT using Western blot. s, l and r, EndoH sensitive, intermediate and resistant, respectively. **(b-c)** Representative immunofluorescent images of LPL with (c) and without (b) co-labeling with KDEL in gonadal WAT. Arrows, tiny LPL puncta formed in WT and Atg7^{AdipCre} WAT.

Taken together, while we cannot exclude the possibility that a small amount of LPL in the ER is degraded by ER-phagy in WT cells under basal state, it is unlikely that ER-phagy plays a major role in LPL maturation in the ER when ERAD is functional. Hence, SEL1L-HRD1 ERAD serves as a principal quality control mechanism for LPL protein folding and maturation in the ER, and when ERAD is compromised, misfolded LPL accumulate and form aggregates in the ER and were degraded by compensatory ER-phagy. In contrast, macro-autophagy may act as a backup system when SEL1L-HRD1 ERAD function is impaired. We have now added a paragraph to discuss the role of

autophagy in the Discussion section of the revised manuscript on page 11 and pasted below.

“Under basal state with functional SEL1L-HRD1 ERAD, our data show that autophagy plays a minor role in the maturation of LPL in the ER. In both WT and Atg7^{AdipCre} WAT, it should be noted that LPL does form small extra-ER puncta, likely representing those in storage vesicles as previously reported. This finding is consistent with previous studies showing that inhibiting macroautophagy increased the amount of LPL reached the cell surface. With that said, we cannot exclude the possibility that a small amount of LPL in the ER is degraded by ER-phagy under basal state. By contrast, when SEL1L-HRD1 ERAD is impaired, autophagy/ER-phagy then plays a major role in clearing misfolded aggregates. While molecular events underlying the formation of ER fragments and subsequent coalescence remain vague, this study uncovers the biochemical chain of events involved in disposing of misfolded protein in the ER. Further characterization of the pathways and mechanisms underlying the formation of ER fragments and cellular inclusions as well as their interaction with various quality control pathways in vivo is key to the understanding of the disease pathogenesis associated with ER protein misfolding, and basis for future therapy.”

Lastly, we have changed the y-axis presentation in Supplementary Fig. 2c to that in Supplementary Fig. 1a, which revealed the similar levels of Xbp1 splicing levels between WT and Atg7^{AdipCre} WAT, pointing to a subtle, if any, ER stress in WAT in the absence of autophagy.

Fig 3b – a better neg control should be found. Even though the quantifications indicate n.s., the chosen blot clearly shows an impact of SEL1-loss on GLUT4. Plus blot for CCPG1.

We thank the reviewer for this great comment. We have now included calnexin as an additional control (Fig. 3e of revised manuscript) and CCPG1 (Response Fig. 4). As we are pursuing CCPG1 in a separate study, we left out CCPG1 in the revised manuscript.

Response Figure 4. Immunoblot analysis of LPL and other proteins in gonadal WAT.

3) Might have missed it – are found KDEL clusters in DKO surrounded by a single membrane (one big phase separated body within the ER/ER fragment) and do several of those accumulate? Please revise description of findings shown in Fig 2. It is also unclear, if you see extended ER-tubules and CERFs.

We thank the reviewer for this great comment. CERFs are the clusters of many ER fragments, which undergo further fusion (Response Fig 5a). Our TEM and 3D FIB-SEM analyses failed to reveal tubular structures in CERFs (Fig. 2g of revised manuscript). We have now clarified this in the text of revised manuscript.

Response Figure 5. CERFs contain membranes. (a) Representative TEM image of CERF in DKO BAT. Arrows, membrane of CERF. (b) Representative immunogold labeling of the ER marker BiP in DKO BAT. Green dot lines, CERFs.

4) Single channels of Fig. 3c, 4g, and 5f should be shown (in main or sup figures). Here a grey scale presentation instead of colored mode should be used to avoid bias. Other single channel presentations of the figures should be also presented in/changed to grey scale.

We thank the reviewer for this great comment. We have made changes as suggested (Response Fig 6).

Response Figure 6. Representative confocal microscopic immunofluorescent images using single channel presentation shown in (a) Supplementary Fig.4a, (b-d) Supplementary Fig. 5d-f, (e) Supplementary Fig. 6a of the revised manuscript.

Minor points:

- Line 62: replace unknown (not true as there are some publications related to this topic) with poorly understood.
- Line 302-303: The authors did not show that SEL1L-HRD1 ERAD regulates ER-phagy. Please remove/adapt this sentence.

We thank the reviewer for these great comments. We have made changes throughout the text per your suggestion.

Reviewer #2:

Remarks to the Author:

The physiological roles of ER-phagy and ERAD, and the intersection between these, is poorly understood. Wu et al provide data using transgenic ERAD-deficient or macroautophagy-deficient mice that the two processes compensate one another in ER proteostasis. In particular, preventing the formation of ER chaperone-delimited LPL protein clusters, composed of multiple fragments of ER, in adipose tissue. The experimental data demonstrating this phenomenon are clear, robust and will be of great interest to the field. I recommend eventual publication. My only caveats are in respect of the framing of the findings.

We thank the reviewer for his/her positive and insightful comments on our work.

These do not necessarily require additional experimentation to address:

1) The CERF compartment wherein the LPL aggregates reside could potentially be another layer of quality control (it for certain indicates a redundancy between autophagy and ERAD in quality control). However, in the absence of a genetic intervention that disrupts formation of this compartment, the authors should not imply a definitive cytoprotective effect of CERF formation.

We thank the reviewer for this great comment with which we agree. We now have revised the text accordingly to tone down the cytoprotective effect of CERF formation.

2) Please clarify: the CERF-unrelated p62 compartments that form in DKO cells. Are these seen in ATG7 deficient tissue or is this just in DKO? If just in DKO, what are these and why do they form in response to combined ERAD and ER-phagy inhibition. Are they a parallel pathway to CERFs?

We thank the reviewer for this great comment. We also observed the P62-containing inclusions/puncta in *Atg7^{AdipCre}* adipose tissues, albeit much smaller in size and much fewer in frequency when compared to those in *DKO* adipose tissue (Response Fig. 7a). We noted a large amount lysosomes and ER surrounding the P62-containing puncta in *Atg7^{AdipCre}* and *DKO* adipose tissues, respectively (arrows, Response Fig. 7b). These findings point to a possible role of SEL1L-HRD1 ERAD in macro-autophagy substrate recruitment and degradation, representing another layer of complexity in ERAD and autophagy crosstalk - an open question that deserves further investigations.

Response Figure 7. Synergism between ERAD and autophagy in the formation of P62 inclusions. (a) Representative immunofluorescent images of P62 in gonadal WAT with quantitation of P62 inclusion size shown on the right. Arrows, P62 inclusions. ****p*<0.001 by Student's *t*-test. (b) Representative TEM images of P62 inclusion formed in *Atg7^{AdipCre}* and *DKO* gonad WAT. White and black arrows, lysosome-like and ER-like structures surrounding the P62 inclusions.

3) Technically, the authors inhibit global macroautophagy not just ER-phagy upon ATG7 knockout. Although I agree the basis of the CERF formation is likely to be defective ER-phagy, the authors should be clear on this limitation of their study within their writing (perhaps in future mouse strain studies ER-phagy could be targeted directly).

We thank the reviewer for this great comment. We have now revised the wording (to be more careful to avoid overstatement of ER-phagy vs. autophagy) and discussed this limitation in the revised manuscript on page 11 and pasted below. ER-phagy is currently difficult to study as there are several potential functionally redundant receptors.

"Our data demonstrate that SEL1L-HRD1 ERAD is a critical regulator of LPL biology by degrading misfolding LPL in the ER. If not cleared efficiently, these LPL forms aggregates in the ER, which are cleared

by selectively autophagy, i.e. ER-phagy. We acknowledge that the use of Atg7 KO mouse model is not ideal to study the ER-phagy process per se. However, despite of our limited understanding of ER-phagy in vivo, we were able to visualize the ER-phagy process specifically with the use of many powerful imaging tools including confocal microscopy, immunogold labeling and TEM. One notable hit in the IP-MS of LPL is the non-canonical ER-phagy receptor CCPG1 – an active investigation in the laboratory.”

In respect of point 3, I note that CCPG1 ER-phagy receptor is detected in the mass spec. Labelling of the CERFs with this or other ER-phagy proteins (FAM134B/C?) could provide some further evidence of the role of ER-phagy. However, I am aware of the severe technical limitations (maybe impossibility) of tissue-level detection of CCPG1 protein, in particular, so don't think it is fair to require this for publication.

We thank the reviewer for this great comment. Indeed, our preliminary data shown in Response Figure 1-2 are really exciting in terms of CCPG1 function in WAT. We are now pursuing this direction as a separate study (please see detailed response above to Reviewer 1). This is the last project of the first author before she heads out for her postdoctoral training.

Reviewer #3:

Remarks to the Author:

The authors have used knockout mice for adipocyte specific gene knockouts for white and brown fat. Sel1L deficient adipocytes were also generated and treated or not with tunicamycin, brefeldin A and cycloheximide. Adipocytes with CRISPR/Cas9 targeted Sel1L and HRD1 from lentivirus transduction were also generated. The above were used to test for compensatory mechanisms in response to impaired SEL1L-HRD1 ERAD. The authors report Xbp1 splicing in wt and SEL1L AdipCre white adipose tissue , cleaved caspase 3 and TUNEL staining, cDNA profiling microarray analysis, LC3-I, LC3-II , P62, and TEM imaging. They report for double KO Sel1L/Atg7, western blots of SEL1L, HRD1, ATG7 in white adipose tissue, size and weight of white fat pads, histology, caspase 3 cleavage and Tunel staining and Xbp1 splicing , immunogold labeling of P62 ,KDEL, and TEM of white and brown adipose tissue. The authors report on proteins characterized in immune-precipitates of SEL1L from brown adipose tissue deleted for HRD1. Lipoprotein lipase was selected for further study with western blots and qPCR reported for HRD1 deleted adipocytes. Immunofluorescence and immunogold labeling and TEM of white adipose tissue of mice deleted for Sel1l and double KO Sel1l/Atg7 is reported as well as serum triglyceride. The authors report as well, proteins characterized in immune-precipitates of lipoprotein lipase from double KO white adipose tissue by mass spectrometry, western blots, immune-fluorescence. The authors also report in vitro phase separation of mCherry lipoprotein lipase wt or mutant, and EGFP-BiP after overnight dialysis in buffer and confocal microscopy, immunofluorescence, FRAP, or immunogold labeling and TEM. The authors conclude that a new quality control compartment (CERF) sequesters protein aggregates in the ER. The authors' conclusion may need a consideration of each of the individual experiments in this manuscript as well as papers published from others over the years concerning quality control compartments.

We thank the reviewer for his/her positive and insightful comments on our work.

For the individual experiments, are the observations in extended data Figures 1a,b,c,d taken from experiments done for ref.28?

*We thank the reviewer for this great comment. Data shown in Supplementary Fig. 1a-c are new experiments done by the first author with additional positive and negative controls and quantitation - distinct from those in the previous study (Sha et al., 2014). Supplementary Fig. 1d was the same array data and processed the same way as that in the previous study (Sha et al., 2014). The only difference is the method used for pathway enrichment analysis – the previous one used the gene set enrichment analysis (GSEA) (Sha et al., 2014); while the current study analyzed enriched KEGG pathways. We have provided detailed analysis procedure in the *Methods* section.*

Have the experiments in Fig1 been done on n=3 biological replicates? For example does n=3 for TEM represent different animals? Why is the magnification for WT in Fig 1b lower than Sel1L adipcre and not indicated for wt. The reported double membranes (mitochondria of course have double membranes normally) or ER like structures are not obvious in Fig1b or extended data Fig 1e. For Fig 1c how was the

quantification done especially since the cells are large and the data are expressed per adipocyte? This is not indicated in the Experimental Procedures. For all Figures with histograms showing means and SEM, are they with biological replicates and could they be indicated with SD that more accurately shows variations in the observations?

We thank the reviewer for these great comments. The experiments in Fig. 1 were performed by using ≥ 3 biologically independent mice, which are indicated in the figure legends of the revised manuscript. In Fig. 1b, scale bars are now included for WT to avoid any confusion. Magnification has been confirmed to be the same for all panels. We have now modified the annotation in Fig. 1b to show the double-membrane

autophagosomes under TEM (Response Fig. 8). We acknowledge that the structure inside of the autophagosomes are difficult to discern, some of which do look like the ER (asterisk, Response Fig. 8). For Fig 1c of the revised manuscript, the quantitation was performed according to equation below, which is now described in the *Methods* section.

Response Figure 8. TEM images of gonadal WAT of *Sel1L^{AdipCre}* mice. The boxed image on the right shows the autophagosome-like structures at high magnification shown on the right. Arrows, the double membrane; *, ER-like structures inside.

$$\# \text{ of AV per adipocyte} = \frac{\text{Total \# of AV identified in adipocytes under TEM}}{\text{Total \# of adipocytes under TEM}}$$

For all the histograms, points represent biological replicates, i.e. one point one mouse/sample – we now have now specified this in the figure legends and the raw data were included all the data points as “source data” in the excel file. We have changed SEM to SD to reflect the variability.

For Figs 2b and extended data Fig 3b again what is claimed in the text is not seen in the images – where are the inner and outer membranes and are they membranes? A minor point is that the mag bars are incorrect at least for one of images in extended data Fig3b.

We thank the reviewer for these great comments. We apologize for the confusion. We have now modified the annotation in Fig. 2 and corresponding text to improve the clarity in the revised manuscript. Of note, in immunogold experiment, no membrane structure is visible (Fig. 2b). Under TEM, P62 inclusion is a membraneless structure, while CERFs are membrane-containing structures (Response Fig. 9). We have confirmed the scale bars in all figures including Supplementary Fig. 3b of the revised manuscript.

Response Figure 9. TEM images of gonadal WAT of *DKO* mice showing membraneless P62 inclusion (#) and membrane (arrows)-containing CERFs (*).

For the IP MS experiments, the reporting is incomplete and unclear. The authors may have used peptide-spectrum matches but this is not defined. Missing are n=3 biological replicates for the IP-MS experiments, the peak lists, search engine, sequence database, enzymes used, false discovery rates at peptide and protein levels as well as what the numbers mean inside the extended data tables 1,2,3. Are these all the proteins characterized in the IP-MS? There are several other additional needed criteria for reporting the authors may consider for mass-spec based data.

We thank the reviewer for these comments. We now have defined PSMs (peptide spectrum matches) in the text and included the relevant information for IP-MS experiments in the *Methods* section of the revised manuscript. For SEL1L IP-MS, we included 3 WT and 3 *Hrd1^{Ucp1Cre}* mice with two independent repeats; for LPL IP-MS, we included WAT pooled from 5 WT, 6 *Sel1L^{AdipCre}*, 3 *Atg7^{AdipCre}* and 7 *DKO* mice. The protein

digestion was performed by trypsin. Proteins were identified by searching the MS/MS data using Proteome Discoverer (v2.4, Thermo Scientific). False discovery rate (FDR) was determined using Percolator and proteins/peptides with a FDR of $\leq 1\%$ were retained for further analysis. The numbers in the extended data tables 1-3 (supplementary table 1-3 of revised manuscript) are (normalized) PSMs. We have specified this in corresponding figure legends, Supplementary tables and the *Methods* section. The proteins listed in the extended data tables 1, 2 and 3 (supplementary table 1-3 of revised manuscript) were identified after applying certain criteria which have been indicated in corresponding figure legends, tables and the *Methods* section of the revised manuscript.

How were the data acquired for Fig3a, what is a hit? What are the clusters in Fig 4a,b,c,d ? A minor point is for the authors to check if they confused possibly the interpretation of the glucuronosyl transferase protein they likely characterized with possibly the calnexin cycle constituent glucosyl transferase that was not? We thank the reviewer for these comments. We have now included more description in the text of the revised manuscript. The criteria for the identification of proteins in IP-MS datasets of Fig. 3a (Fig.3b of revised manuscript) and clusters in Fig. 4a-d (Fig.5a-d of revised manuscript) were now indicated in corresponding figure legends, tables and the *Methods* section of the revised manuscript. In brief, the hits in Fig 3a (Fig.3b of revised manuscript) are the proteins pulled down by SEL1L and have largest PSM increase in *Hrd1^{Ucp1Cre}* brown adipose tissue (BAT) over WT tissue. In Fig. 4 (Fig.5 of revised manuscript), clusters 1 and 2 are proteins pulled down by LPL that are either reduced or increased by at least 4-folds in *DKO* WAT vs. WT WAT. We have now clarified in the text regarding the UDP glucuronosyltransferase family 1 member A6 UGT1A6.

For Fig3d, Fig4h the images are difficult to see as are those in Fig5d, and extended data Fig7a. Each figure and inset in the Figures and extended data figures may be considered by the authors to assess if they really show what is claimed.

We thank the reviewer for this great comment. We have replaced these figures with high-resolution and quality images (Response Fig.10), and have carefully gone through the manuscript to ensure there is no overstatement in the revised manuscript.

Response Figure 10. New images for the paper, (a) Fig. 4f, (b) Fig. 5h, (c) Fig. 6d, (d) Supplementary Fig. 7 of the revised manuscript.

Finally for the quality control compartment, have the authors considered relevance to the publications of the Kopito, Lederkremer, Klumperman, Lippincott-Schwarz, Ferro-Novick labs? We thank the reviewer for this great comment. We have now added more relevant references⁵⁻⁹ in the revised manuscript.

REFERENCES

- 1 Gunn, K. H. *et al.* The structure of helical lipoprotein lipase reveals an unexpected twist in lipase storage. *Proc Natl Acad Sci U S A* **117**, 10254-10264 (2020). <https://doi.org:10.1073/pnas.1916555117>
- 2 Roberts, B. S., Yang, C. Q. & Neher, S. B. Characterization of lipoprotein lipase storage vesicles in 3T3-L1 adipocytes. *J Cell Sci* **135** (2022). <https://doi.org:10.1242/jcs.258734>
- 3 Sundberg, E. L., Deng, Y. & Burd, C. G. Syndecan-1 Mediates Sorting of Soluble Lipoprotein Lipase with Sphingomyelin-Rich Membrane in the Golgi Apparatus. *Dev Cell* **51**, 387-398.e384 (2019). <https://doi.org:10.1016/j.devcel.2019.08.014>
- 4 An, M. *et al.* ULK1 prevents cardiac dysfunction in obesity through autophagy-mediated regulation of lipid metabolism. *Cardiovascular research* **113**, 1137-1147 (2017).
- 5 Ferro-Novick, S., Ruggieri, F. & Brodsky, J. L. ER-phagy, ER homeostasis, and ER quality control: implications for disease. *Trends in biochemical sciences* **46**, 630-639 (2021).
- 6 Johnston, J. A., Illing, M. E. & Kopito, R. R. Cytoplasmic dynein/dynactin mediates the assembly of aggresomes. *Cell motility and the cytoskeleton* **53**, 26-38 (2002).
- 7 Kopito, R. R. Aggresomes, inclusion bodies and protein aggregation. *Trends in cell biology* **10**, 524-530 (2000).
- 8 Lederkremer, G. Z. Glycoprotein folding, quality control and ER-associated degradation. *Current opinion in structural biology* **19**, 515-523 (2009).
- 9 Omari, S. *et al.* Noncanonical autophagy at ER exit sites regulates procollagen turnover. *Proceedings of the National Academy of Sciences* **115**, E10099-E10108 (2018).

REVIEWER COMMENTS

Reviewer #1 has asked to be withdrawn from the review process because of personal reasons.

Reviewer #2 (Remarks to the Author):

I am satisfied the authors have addressed my points. I recommend publication.

Reviewer #3 (Remarks to the Author):

The authors have submitted a rebuttal and revised manuscript. They conclude in the manuscript that a storage compartment they name CERF accumulates misfolded lipoprotein lipase in SEL1L and Atg7 adipocyte specific double knock out mice without triggering ERAD or the UPR.

However several considerations remain that may not have been addressed adequately. The IP mass spec data are incomplete and reported incorrectly. From the supplementary tables, only n=1 biological replicate is reported for SEL1L or LPL IPs. Peptide spectrum matches are shown for the SEL1L IP but not the LPL IP with only ratios indicated that cannot be assessed.

KEGG analyses may be unnecessary especially since these are for experiments done in 2014.

The TEM data remain problematic with images still difficult to interpret and quantification based on single experiments (e.g. Fig 1c). Gold particle immunolabeling appears to have been used qualitatively and not quantitatively but the selected images are clear with the superposition of colored dots. The images indicated in Fig 2 d, e, I, j lack the resolution to conclude ER membrane clustering and fusion.

The extension to studies of LPL misfolding seem reasonable as is the conclusion on NP40 solubility and that this represents a storage compartment for misfolded LPL.

In the rebuttal biological replicates are indicated for several of the Figures and the IP mass spectrometry but this cannot be deduced in the manuscript or supplementary tables. For those with quantification, it seems clear that these may be based on several different experiments as biological replicates. When the authors indicate in the text or legend multiple biological repeats, quantification may be considered for those without such quantification.

The in vitro experiments with BiP and WT and mutant LPLs may be considered for a separate manuscript.

One strength is the immunofluorescence imaging of adipose tissue in the double knock out mice that may be considered for quantification with biological replicates.

A minor point is the extra Fig3D at the end of the manuscript ?

We are very grateful to this reviewer for your insightful comments and sincerely apologize for the confusions. We now have addressed each issue very carefully in the revised manuscript.

Reviewer #3 (Remarks to the Author):

The authors have submitted a rebuttal and revised manuscript. They conclude in the manuscript that a storage compartment they name CERF accumulates misfolded lipoprotein lipase in SEL1L and Atg7 adipocyte specific double knock out mice without triggering ERAD or the UPR. However several considerations remain that may not have been addressed adequately.

The IP mass spec data are incomplete and reported incorrectly. From the supplementary tables, only n=1 biological replicate is reported for SEL1L or LPL IPs.

We thank the reviewer for this comment. For IP-MS, we only did the experiment once as the goal of IP-MS in this study was to identify putative substrates or protein components of CERFs, followed by rigorous validation of the top hits using endogenous IP. For SEL1L IP-MS, BAT from 12-week-old mice (one mouse per genotype) was used. For LPL IP-MS, WAT were pooled for two different experiments - Experiment 1: 5 WT, 6 *Sel1L^{AdipCre}* and 7 *DKO* mice; and Experiment 2: 4 WT and 3 *Atg7^{AdipCre}* mice. We now have provided detailed information in figure legend, method section and supplementary tables.

Peptide spectrum matches are shown for the SEL1L IP but not the LPL IP with only ratios indicated that cannot be assessed.

We thank the reviewer for this comment. Raw PSMs are now shown in the supplementary tables.

KEGG analyses may be unnecessary especially since these are for experiments done in 2014.

We thank the reviewer for this comment. In the previous publication, the pathway analysis data was not complete where autophagy was not included (Response Fig. 1) – hence we were not able to refer to this study. We re-analyzed the dataset with more complete pathways shown in the revised manuscript. We will be happy to take it out if deemed critical.

Response Figure 1. Gene set enrichment analysis (GSEA) of Affymetrix microarray as described in our previous study (Sha et al., Cell Metabolism 2014).

The TEM data remain problematic with images still difficult to interpret and quantification based on single experiments (e.g. Fig 1c).

We thank the reviewer for this comment. We have provided images with the highest quality that currently can be achieved in the field, given how challenging it is for TEM using WAT. The TEM images shown in the manuscript are representative images from at least 2 mice each genotype. In Fig. 1c, we counted autophagic vacuoles from 50 WT and 63 *Sel1L^{AdipCre}* adipocytes from 3 mice each. To the best of our knowledge, we were not able to perform statistical analysis for it.

Gold particle immunolabeling appears to have been used qualitatively and not quantitatively but the selected images are clear with the superposition of colored dots.

We thank the reviewer for this comment. As the results for immunogold labeling provide “yes-or-no” answers, we did not feel the data need to be quantitated – in fact, this type of data is really difficult to be quantitative. Also please see editor’s decision.

The images indicated in Fig 2 d, e, I, j lack the resolution to conclude ER membrane clustering and fusion.

We thank the reviewer for this comment. We have provided images with the highest quality that currently can be achieved in the field, given how challenging it is for TEM using WAT. Although we are comfortable with the quality of our TEM images, we have toned down our conclusion in the revised manuscript in response to reviewer’s comments (pasted below). In addition, to better visualize CERF structure, we now provided a video showing 3D FIB-SEM reconstruction of a CERF in the revised manuscript (Supplementary video 1).

Line 149-153: “Providing further support for the nature of ER fragments, we noted clusters of large ER fragments in DKO adipocytes (Fig. 2i), some of which might have fused at the core (arrows, Fig. 2j and Supplementary Video 1). Taken together, these data suggested that ER fragments may coalesce into a unique cellular architecture in adipocytes lacking SEL1L and ATG7, which we termed as the coalescence of ER fragments (CERFs).”

Legend of Figure 2 i-j (line 840-842): “TEM images showing possible clustering (i) and fusion (j) of ER fragments in DKO BAT (n=3 mice). Arrows, likely fused membranes inside CERFs.”

The extension to studies of LPL misfolding seem reasonable as is the conclusion on NP40 solubility and that this represents a storage compartment for misfolded LPL.

We thank the reviewer for this comment.

In the rebuttal biological replicates are indicated for several of the Figures and the IP mass spectrometry but this cannot be deduced in the manuscript or supplementary tables. For those with quantification, it seems clear that these may be based on several different experiments as biological replicates. When the authors indicate in the text or legend multiple biological repeats, quantification may be considered for those without such quantification.

We thank the reviewer for this comment. We have clarified sample size and repeat numbers as well as other details in the figure legends, methods and supplementary tables of the revised manuscript. We now have performed and included quantitations for all the experiments that require quantitation. However, as the results for some immunolabeling (IF or immunogold) provide “yes-or-no” answers, we did not feel the data need to be quantitated.

The in vitro experiments with BiP and WT and mutant LPLs may be considered for a separate manuscript.

We thank the reviewer for this comment, but please see editor’s decision asking us to keep it in the revised version.

One strength is the immunofluorescence imaging of adipose tissue in the double knock out mice that may be considered for quantification with biological replicates.

We thank the reviewer for this comment. All the IF experiments were repeated with at least two independent samples, but we feel that there was no need for quantitation as the conclusion here is whether there are foci in WAT (green arrows, Fig. 4c-d), or whether LPL is localized to the capillary or inside the cells (e.g. Fig. 4d-e). These are yes or no questions as the difference between the genotypes is black and white.

A minor point is the extra Fig3D at the end of the manuscript ?

We thank the reviewer for this comment. We have fixed this issue in this submission.